# Multifaceted Defects in Monocytes in Different Phases of Chronic Hepatitis B Virus Infection: Lack of Restoration after Antiviral Therapy

Debangana Dey,[a] Sourina Pal,[a] Bidhan Chandra Chakraborty,[b] Ayana Baidya,[a] Soham Bhadra,[a] Ranajoy Ghosh,[c] Soma Banerjee,[a] S. K. Mahiuddin Ahammed,[d] Abhijit Chowdhury,[d] ⓘD Simanti Datta[a]

[a]Centre for Liver Research, School of Digestive and Liver Diseases, Institute of Post Graduate Medical Education and Research, Kolkata, India
[b]Multidisciplinary Research Unit, Institute of Post Graduate Medical Education and Research, Kolkata, India
[c]Division of Pathology, School of Digestive and Liver Diseases, Institute of Post Graduate Medical Education and Research, Kolkata, India
[d]Department of Hepatology, School of Digestive and Liver Diseases, Institute of Post Graduate Medical Education and Research, Kolkata, India

**ABSTRACT** Monocytes play an important role in the control of microbial infection, but monocyte biology during chronic hepatitis B virus (HBV) infection (CHI) remains inadequately studied. We investigated the frequency, phenotype, and functions of monocyte subsets in different phases of CHI, namely, immune tolerance (IT), hepatitis B early antigen (HBeAg)-positive/HBeAg-negative chronic hepatitis B (EP-/EN-CHB, respectively), and inactive carrier (IC), identified factors responsible for their functional alterations, and determined the impact of antiviral therapy on these cells. Flow cytometric analysis indicated that HLA-DR$^+$ CD14$^{++}$ CD16$^-$ classical monocytes were significantly reduced while HLA-DR$^+$ CD14$^{++}$ CD16$^+$ intermediate and HLA-DR$^+$ CD14$^+$ CD16$^{++}$ nonclassical monocytes were expanded in IT and EP-/EN-CHB compared with those in IC and healthy controls (HC). In comparison to IC/HC, monocytes in IT and CHB exhibited diminished expression of Toll-like receptor 2 (TLR-2)/TLR-4/TLR-9 and cytokines interleukin-12 (IL-12)/tumor necrosis factor alpha (TNF-$\alpha$)/IL-6 but produced higher levels of IL-10/transforming growth factor $\beta$ (TGF-$\beta$). Further, monocytes in CHB/IT showed impaired phagocytosis and oxidative response relative to those in IC/HC. *In vitro* assays indicated that high titers of hepatitis B surface antigen (HBsAg) present in IT/CHB and of IL-4 in CHB triggered the functional defects in monocytes via induction of $\beta$-catenin. Additionally, monocyte-derived M1 macrophages of CHB/IT produced fewer proinflammatory and more anti-inflammatory cytokines than those of IC/HC, while in CHB/IT, the monocytes skewed the differentiation of CD4$^+$ T cells more toward regulatory T cells and a Th2 phenotype. Moreover, monocytes in CHB and IT overexpressed chemokine receptor CCR2, which coincided with increased intrahepatic accumulation of $\beta$-catenin$^+$ CD14$^+$ cells. One year of tenofovir therapy failed to normalize monocyte functions or reduce serum HBsAg/IL-4 levels. Taken together, monocytes are functionally perturbed mostly in IT and EP-/EN-CHB phases. Targeting intramonocytic $\beta$-catenin or reducing HBsAg/IL-4 levels might restore monocyte function and facilitate viral clearance.

**IMPORTANCE** Chronic HBV infection (CHI) is a major cause of end-stage liver disease for which pharmacological treatments currently available are inadequate. Chronically HBV-infected patients fail to mount an efficient immune response to the virus, impeding viral clearance and recovery from hepatitis. Monocytes represent a central part of innate immunity, but a comprehensive understanding on monocyte involvement in CHI is still lacking. We here report a multitude of defects in monocytes in chronically HBV-infected patients that include alteration in subset distribution, Toll-like receptor expression, cytokine production, phagocytic activity, oxidative response, migratory ability, polarization of monocyte-derived macrophages, and monocyte–T-cell interaction. We demonstrated that high levels of hepatitis B virus surface antigen and IL-4 potentiate these defects in monocytes via $\beta$-catenin induction while therapy

Address correspondence to Simanti Datta, seemdatt@gmail.com.
The authors declare no conflict of interest.

with the nucleotide analog tenofovir fails to restore monocyte function. Our findings add to the continuing effort to devise new immunotherapeutic strategies that could reverse the immune defects in CHI.

**KEYWORDS** monocyte subsets, hepatitis B virus surface antigen, IL-4, $\beta$-catenin, tenofovir, Treg, M2 macrophages

The outcome of chronic hepatitis B virus (HBV) infection (CHI) and the pathogenesis of liver disease are largely determined by immune-mediated host-virus interactions (1, 2). Inability to achieve sustained viral control in CHI has been correlated with the incapacity of the host to evoke an effective immune response against the virus (2). Monocytes represent a critical component of innate immunity that plays a fundamental role in the control of microbial infection and also contributes to the pathogenesis of inflammatory diseases (3). They are equipped with a set of scavenger and Toll-like receptors (TLRs), which can recognize pathogen-associated molecular patterns (PAMPs) and trigger intracellular signaling cascades leading to the expression of a variety of proinflammatory cytokines, enhanced phagocytic activity, and generation of reactive oxygen and nitrogen intermediates (3, 4). These events orchestrate the early host response to infection that promotes the clearance of the pathogen. Based on the expression of CD14 and CD16, human monocytes can be divided into three subsets, $CD14^{++}/CD16^{-}$ (classical), $CD14^{++}/CD16^{+}$ (intermediate), and $CD14^{+}/CD16^{++}$ (nonclassical), whose relative percentages and biological functions had been reported to vary in different diseases (5). The classical monocytes are the most abundant subset, mainly primed for phagocytosis, innate sensing/immune responses, and migration. The nonclassical monocytes are specialized in complement- and Fc-gamma-mediated phagocytosis and adhesion and also exhibit a heightened response to viruses by producing proinflammatory cytokines. The intermediate monocytes, on the other hand, are well suited for antigen presentation, cytokine secretion, and apoptosis regulation (5). The monocytes traffic to the site of infection/inflammation, and depending on microenvironmental stimuli, they differentiate into either M1 macrophages having a pronounced proinflammatory phenotype or M2 macrophages with anti-inflammatory attributes (3). In addition, studies have indicated that the monocytes have a fundamental role in the presentation of antigens to cognate T cells and in $CD4^{+}$ T-cell differentiation into distinct effector cells that could impact the elimination of microbes and disease pathogenesis (6). Relatively little is known about the effects of chronic viral infection on monocytes. Monocytes/macrophages serve as an important reservoir of HIV, and the disease progression has been found to be closely linked to the expansion of $CD16^{+}$ monocytes (7). Altered TLR signaling and cytokine production by monocytes had been described in hepatitis C virus (HCV)-infected patients (8). In chronically HBV-infected patients, changes were observed in the frequencies of monocyte subsets and regulation of TLR expression by HBV precore and surface protein had been reported (9, 10). However, there is still a lack of comprehensive understanding of monocyte biology during CHI, whose natural history includes four dynamic phases, namely, (i) immune tolerance (IT) (recently renamed hepatitis B early antigen [HBeAg]-positive chronic HBV infection), (ii) HBeAg-positive chronic hepatitis B (EP-CHB), (iii) inactive carrier (IC) (also termed HBeAg-negative chronic HBV infection), and (4) HBeAg-negative chronic hepatitis B (EN-CHB) (11). Hence, the present study aimed to appraise the distinct phenotypes and functions of monocytes in different phases of CHI, identify the viral and host factors that could modulate the properties of these cells, and study the interaction of monocytes with $CD4^{+}$ T cells. We also explored whether therapy with the nucleotide analog tenofovir could restore the functional capacity of the monocytes in CHB patients. This integrated knowledge would be vital for the designing of more effective therapies for CHB.

## RESULTS

**Clinical, serological, and demographic data.** A total of 65 chronically HBV-infected patients were categorized as IT ($n = 10$), EP-CHB ($n = 15$), IC ($n = 22$), and EN-CHB ($n = 18$) based on clinical, serological, virological, and histological assessments (see Table S1 in the supplemental material). In addition, 19 healthy controls (HC) were included.

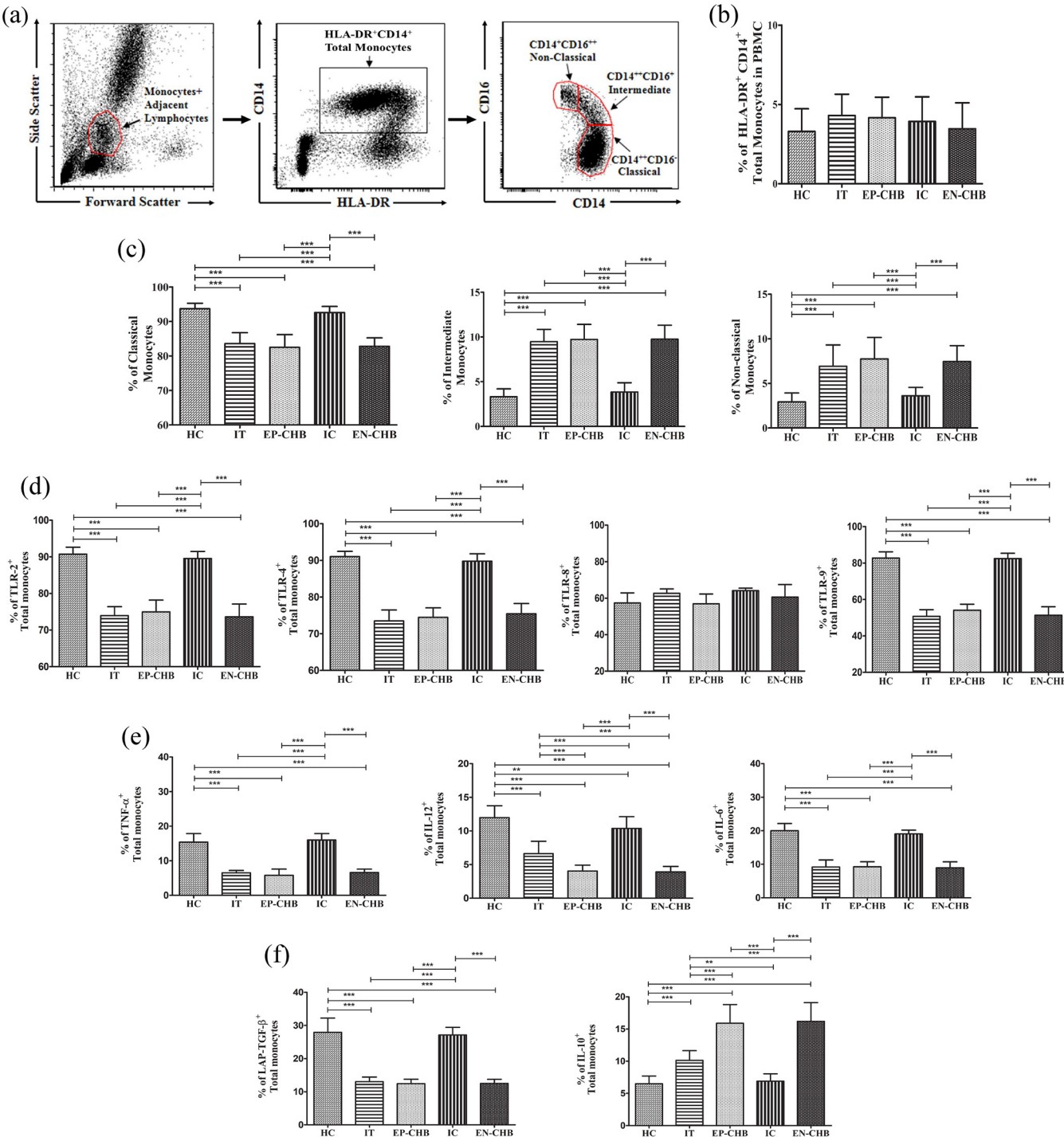

**FIG 1** Frequency, phenotype, and function of monocytes in different phases of chronic HBV infection. (a) Sequential gating strategy for identification of HLA-DR$^+$ CD14$^+$ total, HLA-DR$^+$ CD14$^{++}$ CD16$^-$ classical, HLA-DR$^+$ CD14$^{++}$ CD16$^+$ intermediate, and HLA-DR$^+$ CD14$^+$ CD16$^{++}$ nonclassical monocytes using immunophenotyping followed by flow cytometry. (b and c) The relative abundance of total (b) and classical, intermediate, and nonclassical (c) monocytes in immune tolerance (IT) ($n = 10$), HBeAg-positive chronic hepatitis B (EP-CHB) ($n = 15$), inactive carrier (IC) phase ($n = 22$), HBeAg-negative CHB (EN-CHB) ($n = 18$), and healthy controls (HC) ($n = 19$). (d to f) Percentages of total monocytes expressing TLR-2, TLR-4, TLR-8, and TLR-9 (d), TNF-$\alpha$, IL-12, and IL-6 (e), and LAP-TGF-$\beta$ and IL-10 (f) in different study groups as determined by flow cytometric analysis. Means ± standard deviations are given. Means among groups were compared by one-way ANOVA with Tukey's multiple-comparison test (**, $P < 0.005$, and ***, $P < 0.0001$).

**Variability in the distribution of monocyte subsets in different phases of CHI.** We first determined the frequency of total circulating monocytes and their subsets in different phases of CHI (Fig. 1a to c). The frequencies of HLA-DR$^+$ CD14$^+$ total monocytes were comparable across all study groups (Fig. 1b). However, the proportion of HLA-DR$^+$ CD14$^{++}$ CD16$^-$

(classical) monocytes was significantly diminished in IT (83.6% $\pm$ 3.1%), EP-CHB (82.5% $\pm$ 3.6%), and EN-CHB (82.7% $\pm$ 2.4%) relative to IC (92.5% $\pm$ 1.7%) and HC (93.6% $\pm$ 1.6%), while a stark increase was observed in HLA-DR$^+$ CD14$^{++}$ CD16$^+$ (intermediate) and HLA-DR$^+$ CD14$^+$ CD16$^{++}$ (nonclassical) monocytes in IT (intermediate/ nonclassical monocytes; 9.4% $\pm$ 1.3%/ 6.9% $\pm$ 2.3%), EP-CHB (9.7% $\pm$ 1.6%/ 7.7% $\pm$ 2.4%), and EN-CHB (9.7% $\pm$ 1.5%/ 7.4% $\pm$ 1.7%) compared to IC (3.8% $\pm$ 1.0%/ 3.6% $\pm$ 0.9%) and HC (3.3% $\pm$ 0.8%/ 2.9% $\pm$ 0.9%, respectively) (Fig. 1c).

**Reduced expression of TLRs and aberrant cytokine profile of monocytes during CHI.** The recognition of PAMPs by TLRs of monocytes represents a critical step for the clearance of infecting microbes. We noted a significant reduction in frequencies of total monocytes expressing TLR-2, TLR-4, and TLR-9 in IT and EP-/EN-CHB compared with those in IC and HC while no difference could be perceived in the incidence of TLR-8$^+$ monocytes across the groups (Fig. 1d). All three monocyte subsets of IT and EP-/EN-CHB exhibited the declining trends in TLR-2/-4/-9 expression (Fig. S1a). TLR-2 and TLR-4 were equivalently expressed by all subsets whereas TLR-9 was expressed predominantly by the classical monocytes and TLR-8 by both classical and intermediate subsets.

We observed a marked diminution in TLR-regulated proinflammatory cytokines tumor necrosis factor alpha (TNF-$\alpha$), interleukin-12 (IL-12), and IL-6 by total monocytes (Fig. 1e) and all the subsets (Fig. S1b) in IT and EP-/EN-CHB compared with IC or HC. While the expression levels of intramonocytic TNF-$\alpha$ and IL-6 were equivalent in IT and EP-/EN-CHB, IL-12 was significantly lower in EP-/EN-CHB patients than in IT. Further, the frequencies of IL-12$^+$ monocytes were lower in IC than in HC (Fig. 1e). The intermediate and nonclassical monocytes were the major sources of TNF-$\alpha$ and IL-12 whereas higher IL-6 production was detected in classical monocytes (Fig. S1b).

In addition, we evaluated the production of inhibitory cytokines transforming growth factor $\beta$ (TGF-$\beta$) and IL-10 by the monocytes in different phases of CHI. We assessed the expression of the latent form of TGF-$\beta$ in which TGF-$\beta$ remains bound to the latency-associated peptide (LAP) (12), such that lower cell surface LAP expression correlates with higher levels of TGF-$\beta$ secretion. IT as well as EP-/EN-CHB patients displayed a significant decline in the proportion of LAP-TGF-$\beta$-positive monocytes, indicative of raised functional TGF-$\beta$ levels in these patients, compared with IC/HC (Fig. 1f; Fig. S1c). Classical monocytes expressed the lowest LAP-TGF-$\beta$ level in all groups, implying higher secretion of active TGF-$\beta$ (Fig. S1c). Furthermore, an analogous expansion in IL-10$^+$ monocytes (Fig. 1f; Fig. S1c) was apparent in IT and EP-/EN-CHB relative to IC or HC. Notably, monocytes of EP-/EN-CHB showed enhanced IL-10 expression compared with that of IT (Fig. 1f). However, in each phase, the individual subsets displayed similar capacity to produce IL-10 (Fig. S1c).

**Diminished phagocytic activity and oxidative response by monocytes in IT/CHB.** To gain insight into the phagocytic ability of monocytes in chronically HBV-infected patients, we first analyzed on circulating monocytes the expression of Fc$\gamma$RI/CD64, the primary receptor for opsonic uptake of antigens. In comparison to IC and HC, a deficit in phagocytic function of monocytes was noted in IT and EP-/EN-CHB, as evident from the significantly low expression of CD64 in total monocytes, including all subsets (Fig. 2a; Fig. S2a). Classical monocytes exhibited the most prominent expression of CD64 in all disease phases, implying their greater aptitude for phagocytosis than that of other subsets (Fig. S2a). Consistent with decreased CD64 expression, the ability of the monocytes to engulf fluorescein isothiocyanate (FITC)-zymosan particles was also substantially reduced in IT and EP-/EN-CHB compared with that of IC and HC (Fig. 2a; Fig. S2b and c).

Phagocytosis leads to the generation of reactive oxygen and nitrogen species (ROS/RNS, respectively) within the monocytes. We detected the intracellular ROS production in monocytes using a dichlorodihydrofluorescein diacetate (DCFH-DA) probe, which could be oxidized by ROS to a green fluorescent product, dichlorofluorescein (DCF). As inferred from decreasing DCF fluorescence, the capacity of the monocytes (including all subsets) to produce ROS was significantly attenuated in IT and EP-/EN-CHB relative to HC and IC (Fig. 2b; Fig. S3a and b). RNS production is dependent upon nitric oxide that is generated by inducible nitric oxide synthase (iNOS). Compared to IC/HC, iNOS$^+$ total monocytes and all monocyte subsets were reduced in IT and EP-/EN-CHB, suggesting a decrease in iNOS-mediated RNS production

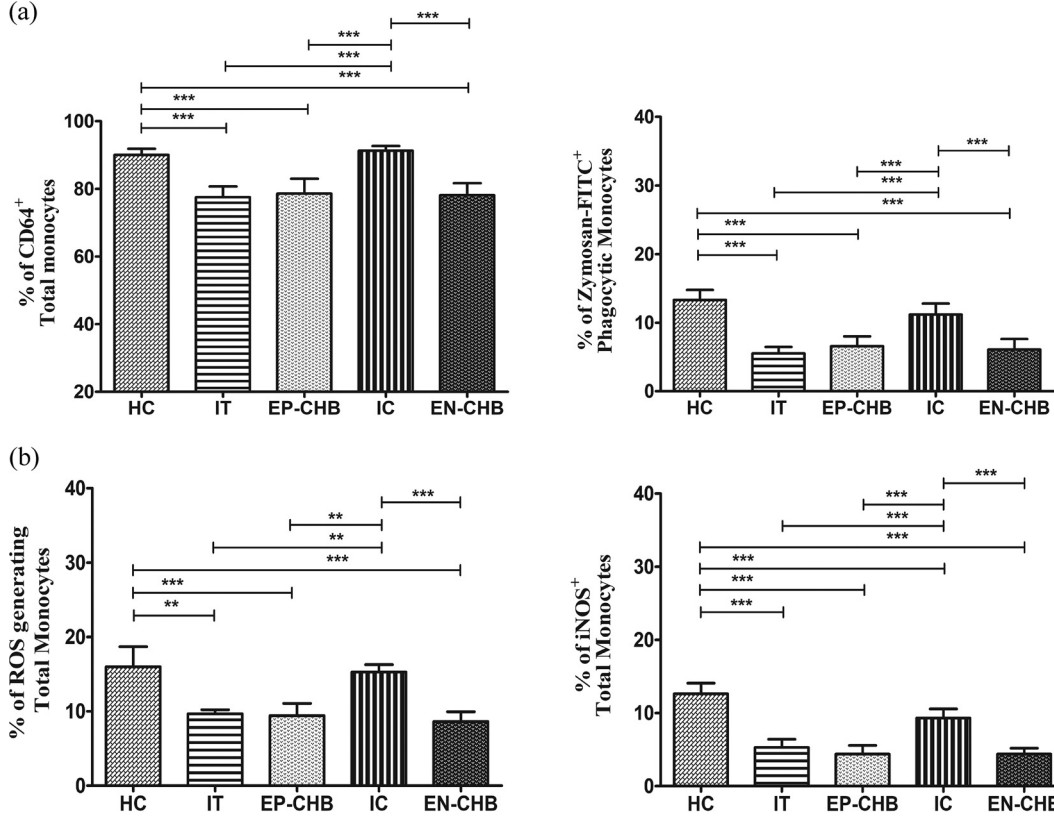

**FIG 2** Phagocytosis and oxidative responses by monocytes during chronic HBV infection. (a and b) Bar diagrams demonstrating frequencies of CD64-expressing monocytes and those with phagocytosed zymosan-FITC particles (a) and ROS-generating and iNOS$^+$ monocytes in immune tolerance (IT) ($n = 10$), HBeAg-positive chronic hepatitis B (EP-CHB) ($n = 15$), inactive carrier (IC) phase ($n = 22$), HBeAg-negative CHB (EN-CHB) ($n = 18$), and healthy controls (HC) ($n = 19$) (b). The expression of CD64 and iNOS was determined on monocytes by immunophenotyping and intracellular assay, respectively. Phagocytosis of the monocytes was determined by the engulfment of FITC-zymosan particles and detection of the reporter signal by flow cytometry. Generation of ROS was determined by oxidation of DCFH-DA to DCF following detection of DCF fluorescence by flow cytometry. Means $\pm$ standard deviations are given. Means among groups were compared by one-way ANOVA with Tukey's multiple-comparison test (**, $P < 0.005$, and ***, $P < 0.0001$).

(Fig. 2b; Fig. S3c). Additionally, IC harbored a significantly lower frequency of iNOS-expressing monocytes than did HC. In all cases, the nonclassical and intermediate monocytes produced higher levels of ROS and expressed higher levels of iNOS than the classical subset (Fig. S3a to c).

**HBsAg and cytokine milieu regulate the monocyte functions in CHI.** We next sought to identify the viral antigen and systemic cytokines that might contribute to the altered monocyte functions during CHI. We first evaluated the serum concentration of HBsAg, the most abundant viral protein in the blood of HBV-infected individuals. The HBsAg level was found to be markedly higher in IT (5.2 $\pm$ 0.9 log$_{10}$ IU/mL) and EP-/EN-CHB (5.3 $\pm$ 0.9 log$_{10}$ IU/mL) than in IC (3.2 $\pm$ 0.5 log$_{10}$ IU/mL) (Fig. 3a). HBsAg titers were found to be inversely correlated with the frequencies of TLR2$^+$, IL-12$^+$, CD64$^+$, and iNOS$^+$ monocytes, while a significant positive correlation was observed between serum HBsAg levels and IL-10-expressing monocytes, implying a potential role of HBsAg in monocyte dysfunction (Fig. 3b). Further, treatment of CD14$^+$ monocytes, sorted from HC, with a high concentration of recombinant HBsAg (rHBsAg) resulted in reduced frequency of classical monocytes and amplification of the nonclassical and intermediate subsets, along with suppression of TLR-2, CD64, IL-12, and iNOS and augmentation of IL-10 expression compared to untreated and $\beta$-galactosidase ($\beta$-Gal)-treated cells (Fig. 3c to h; Fig. S4). On the other hand, even at a low HBsAg concentration, the monocytes exhibited a significant decrease in IL-12 and iNOS expression compared with control setups (Fig. 3e and h).

Despite the similar HBsAg levels in IT and CHB, we noticed a decline in IL-12$^+$ and an increase in IL-10$^+$ monocytes in CHB compared to IT. We postulated that these functional

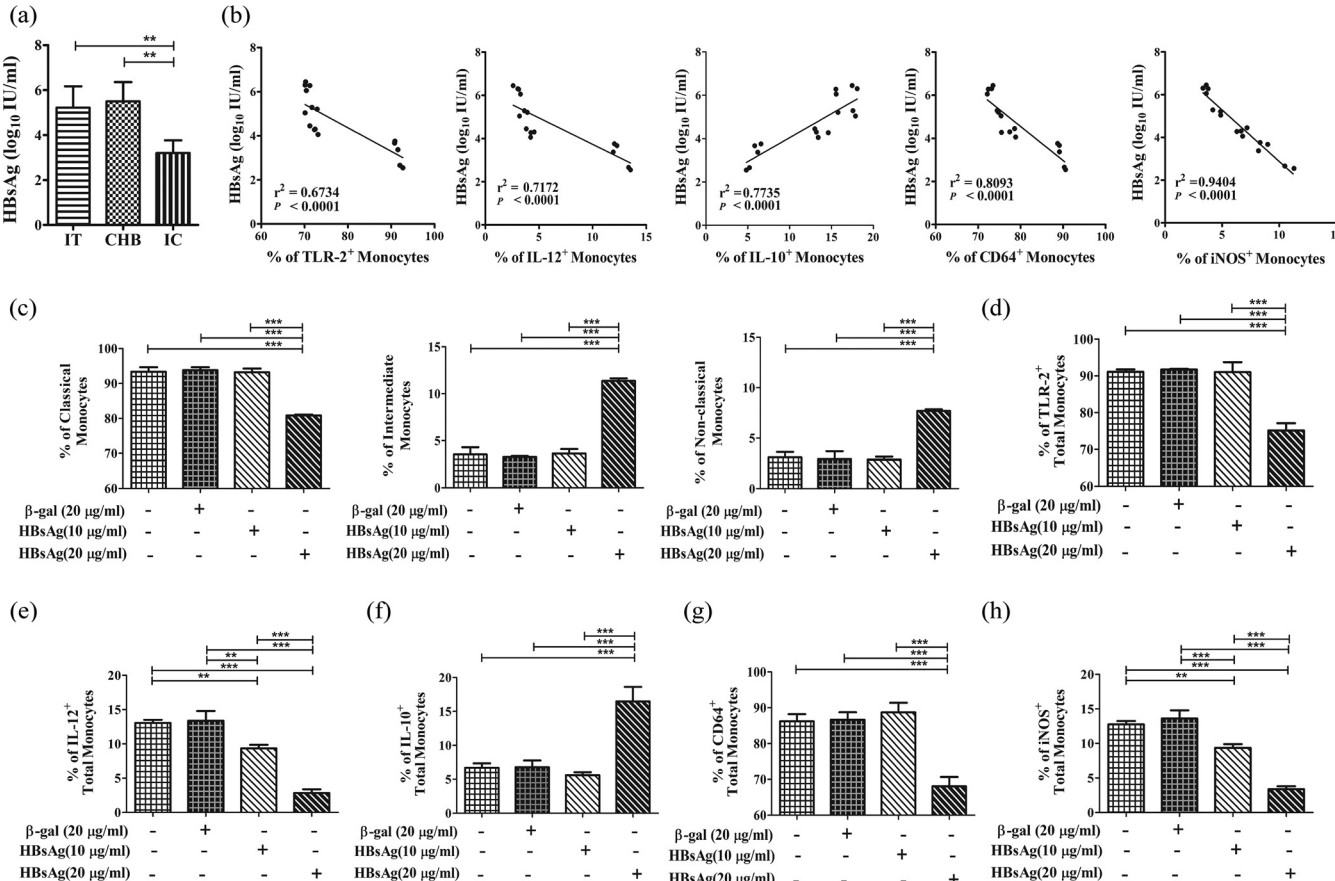

**FIG 3** Effect of viral factor on functional properties of monocytes. (a) Serum levels of hepatitis B virus surface antigen (HBsAg) in different groups of chronically HBV-infected patients quantified on an Abbott Architect 1000sr platform. Means ± standard deviations are given. Statistical significance was assessed by one-way ANOVA followed by Tukey's multiple-comparison test (**, $P < 0.005$). (b) Correlation analysis between percentages of TLR-2-, IL-12-, IL-10-, CD64-, and iNOS-expressing monocytes and level of HBsAg in chronically HBV-infected patients. Correlation was assessed by linear regression analysis. (c to h) The proportion (percentage) of classical, intermediate, and nonclassical monocytes (c) and the expression of TLR-2 (d), IL-12 (e), IL-10 (f), CD64 (g), and iNOS (h) in sorted CD14+ monocytes of healthy controls (HC) treated with or without β-galactosidase (β-Gal) (20 μg/mL) and recombinant hepatitis B virus surface antigen (rHBsAg) (10 μg/mL and 20 μg/mL) for 48 h as detected by flow cytometry. Means ± standard deviations from three individual sets of experiments are given. Statistical significance was assessed by one-way ANOVA followed by Tukey's multiple-comparison test (**, $P < 0.005$, and ***, $P < 0.0001$).

variabilities could be related to the differences in local cytokines in these two phases. Significant increases were found for serum IL-4 and TNF-$\alpha$ exclusively in EP-/EN-CHB phases compared with the other groups (Fig. 4a and b). In addition, we observed that treatment of monocytes with high concentrations of rIL-4 conferred significant diminution in IL-12+ and enhancement in the percentages of IL-10+ monocytes relative to untreated cells, while no discernible change was noted upon rTNF-$\alpha$-treatment (Fig. 4c and d; Fig. S5a and b).

**HBsAg and IL-4 activated $\beta$-catenin in monocytes.** We next investigated the mechanism underlying HBsAg- or IL-4-mediated alteration in the properties of monocytes. Given that $\beta$-catenin could suppress TLR-triggered proinflammatory responses and induce anti-inflammatory cytokines (13, 14), we speculated that HBsAg or IL-4 could promote the aberrant monocyte function through activation of $\beta$-catenin. To test this hypothesis, we first examined the expression of $\beta$-catenin in CD14+ monocytes sorted from HC following separate treatment with exogenous HBsAg and IL-4. It was observed that HBsAg resulted in an ~3.8-fold increase in the $\beta$-catenin protein level on the monocytes of HC (Fig. 4e; Fig. S6a). This induction was, however, not accompanied by an increase in $\beta$-catenin mRNA amounts. Rather, HBsAg-treated monocytes exhibited an ~1.4-fold decline in $\beta$-catenin mRNA abundance (Fig. S6b). These observations suggested that the induction of $\beta$-catenin expression was mainly at the level of protein synthesis or stability. In parallel, $\beta$-catenin+ monocytes were significantly elevated in HBsAg-positive chronically HBV-infected patients compared with HC (Fig. 4f). HBsAg titers correlated positively with percentages of $\beta$-catenin+ monocytes,

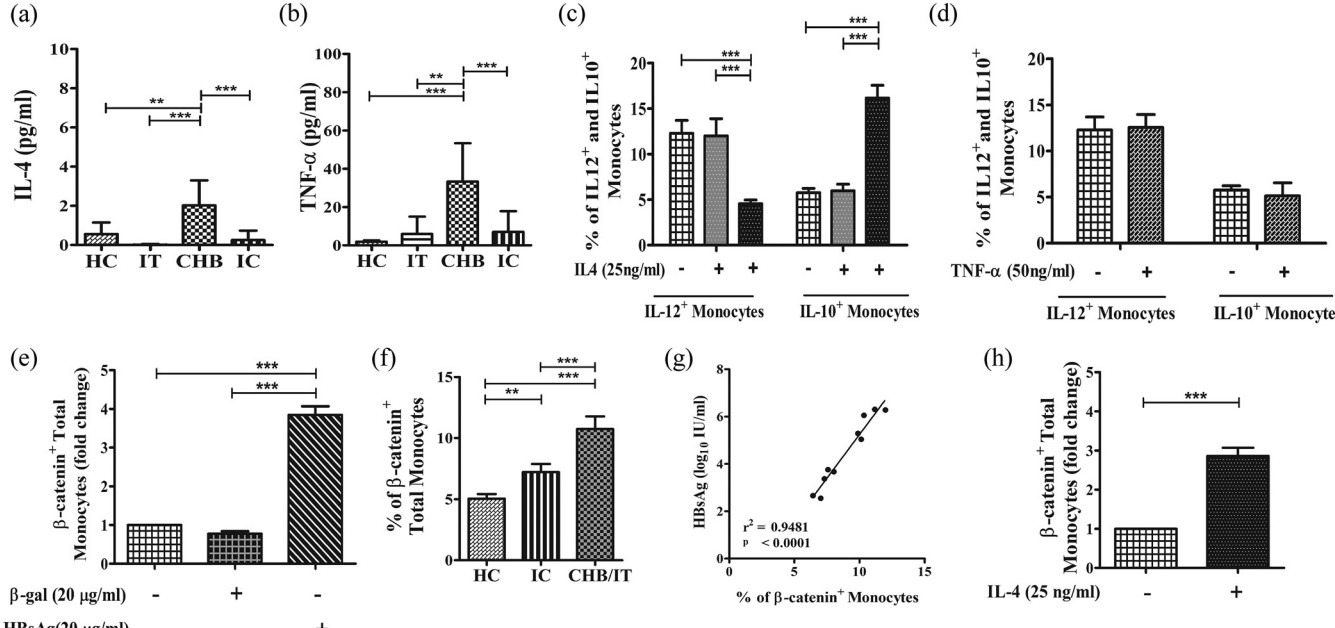

**FIG 4** Effect of host factors on functional properties of monocytes. (a and b) Concentrations in serum of IL-4 (a) and TNF-$\alpha$ (b) in immune tolerance (IT) ($n = 6$), chronic hepatitis B (CHB) ($n = 20$), inactive carrier (IC) phase ($n = 15$), and healthy controls ($n = 10$) quantified by cytokine array using flow cytometry. Means $\pm$ standard deviations are given. Statistical significance was assessed by one-way ANOVA followed by Tukey's multiple-comparison test (**, $P < 0.005$, and ***, $P < 0.0001$). (c and d) Grouped bar diagram of relative percentages of HLA-DR$^+$ CD14$^+$ monocytes expressing IL-12 and IL-10 in PBMC of HC treated with or without rIL4 (5 ng/mL or 25 ng/mL) (c) and TNF-$\alpha$ (50 ng/mL) (d) for 3 days as determined by flow cytometric analysis. (e) Frequency of HLA-DR$^+$ CD14$^+$ $\beta$-catenin$^+$ cells after treatment of sorted CD14$^+$ monocytes of healthy controls (HC) with $\beta$-galactosidase ($\beta$-Gal) (20 $\mu$g/mL) and rHBsAg (20 $\mu$g/mL). Statistical significance was assessed by one-way ANOVA followed by Tukey's multiple-comparison test and two-way ANOVA (**, $P < 0.005$, and ***, $P < 0.0001$), respectively. (f) Bar diagram demonstrating $\beta$-catenin$^+$ total monocytes of HC and patients in different phases of chronic HBV infection. (g) Correlation analysis between percentages of $\beta$-catenin-expressing monocytes and level of HBsAg in chronically HBV-infected patients. Correlation was assessed by linear regression analysis. (h) Frequency of HLA-DR$^+$ CD14$^+$ $\beta$-catenin$^+$ cells following treatment of PBMC of HC with rIL-4 (25 ng/mL). Means $\pm$ standard deviations from three individual sets of experiments are given. Statistical significance was assessed using paired $t$ test (***, $P < 0.0001$).

such that monocytes in IT/CHB, with high HBsAg levels, showed heightened $\beta$-catenin expression compared with those from IC, where the HBsAg level was lower (Fig. 4g). On the other hand, treatment with IL-4 resulted in a ~2.9-fold increase in $\beta$-catenin protein amounts in monocytes of HC, while no change in the mRNA level was noted (Fig. 4h; Fig. S7a and b).

**$\beta$-Catenin/T-cell factor (TCF) inhibitor restored monocyte functions.** To establish the influence of increased $\beta$-catenin in alteration of monocyte functions, we treated sorted CD14$^+$ monocytes of HC with $\beta$-catenin/TCF inhibitor along with rHBsAg/IL-4. Addition of $\beta$-catenin/TCF inhibitor led to enhanced frequency of classical monocytes as well as that of TLR-2$^+$, IL-12$^+$, CD64$^+$, and iNOS$^+$ monocytes but caused reductions in intermediate and nonclassical subsets along with IL-10-expressing monocytes, relative to that observed where rHBsAg or rIL-4 alone was present (Fig. 5a to h; Fig. S4 and S5a). Collectively, these findings signify that the functions of monocytes were compromised by induction of $\beta$-catenin by HBsAg or IL-4.

**_In vitro_-differentiated macrophages from circulating monocytes showed altered cytokine secretion profile during CHI.** We studied the features of monocyte-derived macrophages (MDMs) in chronically HBV-infected patients by first differentiating the peripheral blood monocytes into macrophage-like cells with macrophage colony-stimulating factor (M-CSF) and then generating the M1 or M2 macrophage phenotype by stimulating with either a combination of gamma interferon (IFN-$\gamma$) and bacterial lipopolysaccharide (LPS) or IL-4. Monocyte-derived macrophages (MDMs) were identified as HLA-DR$^+$ CD14$^+$ CD68$^+$ cells (Fig. 6a), and levels of different intracellular cytokines were assessed on these cells. We noted that M1 macrophages, particularly in CHB and also IT, were characterized by marked decline in IL-12 and TNF-$\alpha$ production relative to HC/IC (Fig. 6b). In addition, a heightened frequency of IL-10-expressing M1 macrophages was perceived in CHB followed by IT, while the frequency was much lower in IC and HC (Fig. 6b). M2 macrophages from CHB and IT displayed superior abilities to produce IL-10 compared with IC and HC, although CHB showed

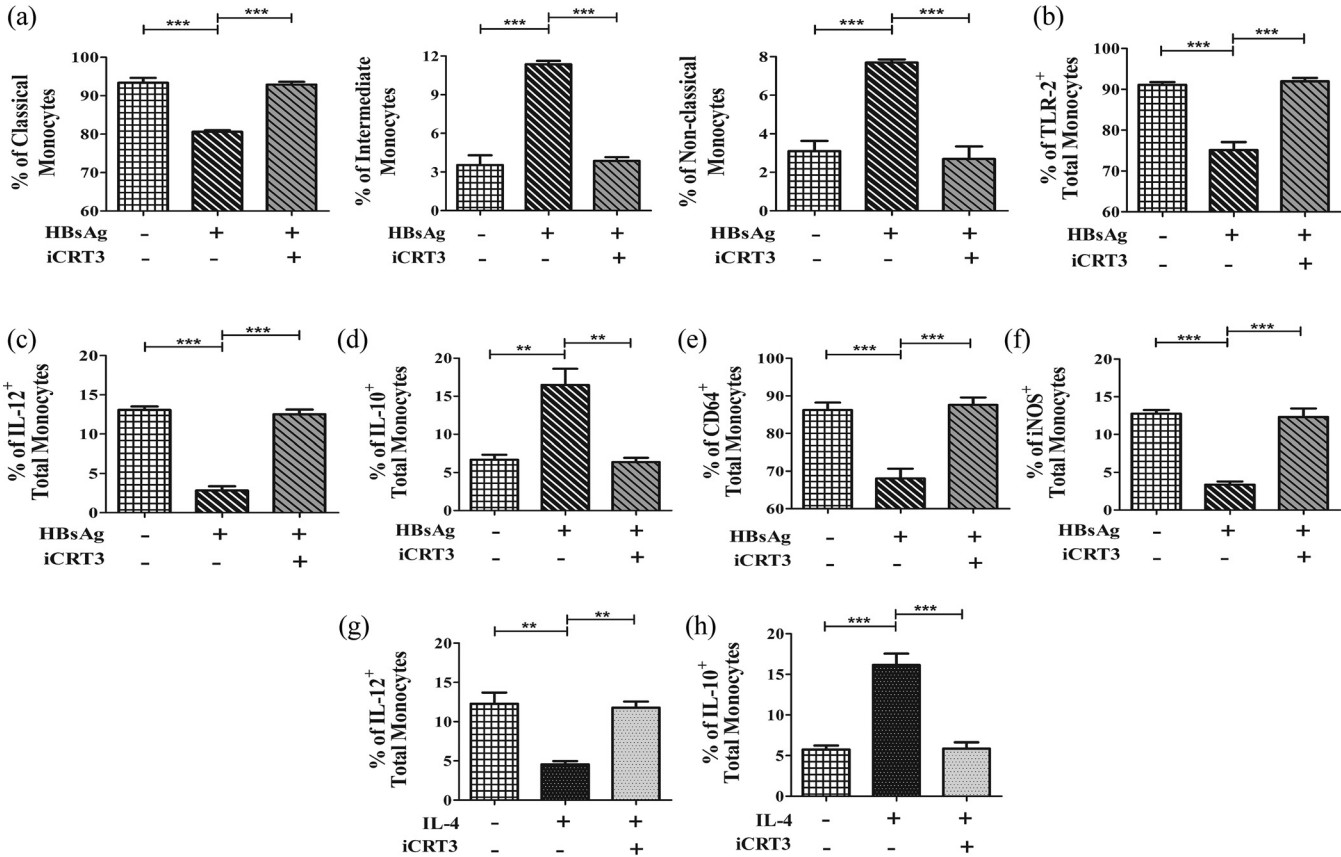

**FIG 5** Impact of $\beta$-catenin on monocyte function. (a to f) Grouped bar diagrams showing proportion (percentages) of classical, intermediate, and nonclassical subsets (a) and expression of TLR-2 (b), IL-12 (c), IL-10 (d), CD64 (e), and iNOS (f) on monocytes treated with rHBsAg (20 $\mu$g/mL) or a combination of rHBsAg (20 $\mu$g/mL) and $\beta$-catenin/TCF inhibitor (iCRT3) (25 $\mu$M). (g and h) Percentages of IL-12$^+$ (g) and IL-10$^+$ (h) monocytes following treatment with rIL-4 (25 ng/mL) or a combination of rIL-4 (25 ng/mL) with iCRT3 (25 $\mu$M). Means ± standard deviations from three individual sets of experiments are given. Statistical significance was assessed by repeated-measures ANOVA (**, $P < 0.005$, and ***, $P < 0.0001$).

significantly higher expression of IL-10 than did IT (Fig. 6c). Irrespective of the study groups, all M2 macrophages expressed little IL-12 and TNF-$\alpha$.

**Monocytes in different phases of CHI dictate distinct CD4$^+$ T-cell differentiation.** Monocytes are known to drive the differentiation of CD4$^+$ T cells into distinct functional populations (6). Given the differential expression of key cytokines by monocytes in chronically HBV-infected patients, we explored whether the monocytes contribute uniquely in promoting any specific CD4$^+$ T-cell differentiation programs in the setting of CHI. Coculture of sorted CD14$^+$ monocytes from CHB and IT with autologous anti-CD3/anti-CD28-stimulated monocyte-depleted peripheral blood mononuclear cells (PBMC) resulted in enrichment of the CD4$^+$ CD25$^+$ FOXP3$^+$ T regulatory cells (Treg) by ~4.4- and ~4-fold, respectively, and an expansion of the CD4$^+$ CCR4$^+$ CCR6$^-$ Th2 subset by ~2.6- and ~2-fold, respectively. However, an equivalent population of Treg or Th2 cells did not emerge in the presence of monocytes sorted from HC/IC. In addition, the monocytes of CHB patients resulted in about ~2.7-fold induction of the CD4$^+$ CCR4$^+$ CCR6$^+$ Th17 subset but had little effect on the differentiation of CD4$^+$ CXCR3$^+$ Th1 cells. Conversely, in IC and HC, monocytes aided in the amplification of Th1 cells by ~4-fold, whereas in IT, a ~2.4-fold rise in Th1 cells was seen. However, no enhancement in monocyte-dependent Th17 cells was perceived in IT, IC, and HC (Fig. 6d).

**Increased mobility traits of monocyte subsets in CHB and IT.** C-C chemokine receptor 2 (CCR2) plays a key role in the recruitment of monocytes into the liver (15). CCR2$^{hi}$ monocytes were found to be markedly elevated in both EP- and EN-CHB patients compared with other study groups, while IT showed a greater percentage of CCR2$^{hi}$ monocytes than did IC/HC (Fig. 7a). This suggests a greater potential of the monocytes in CHB as well as IT to home to the liver. CCR2 was highly expressed by all three monocyte subsets in CHB and IT

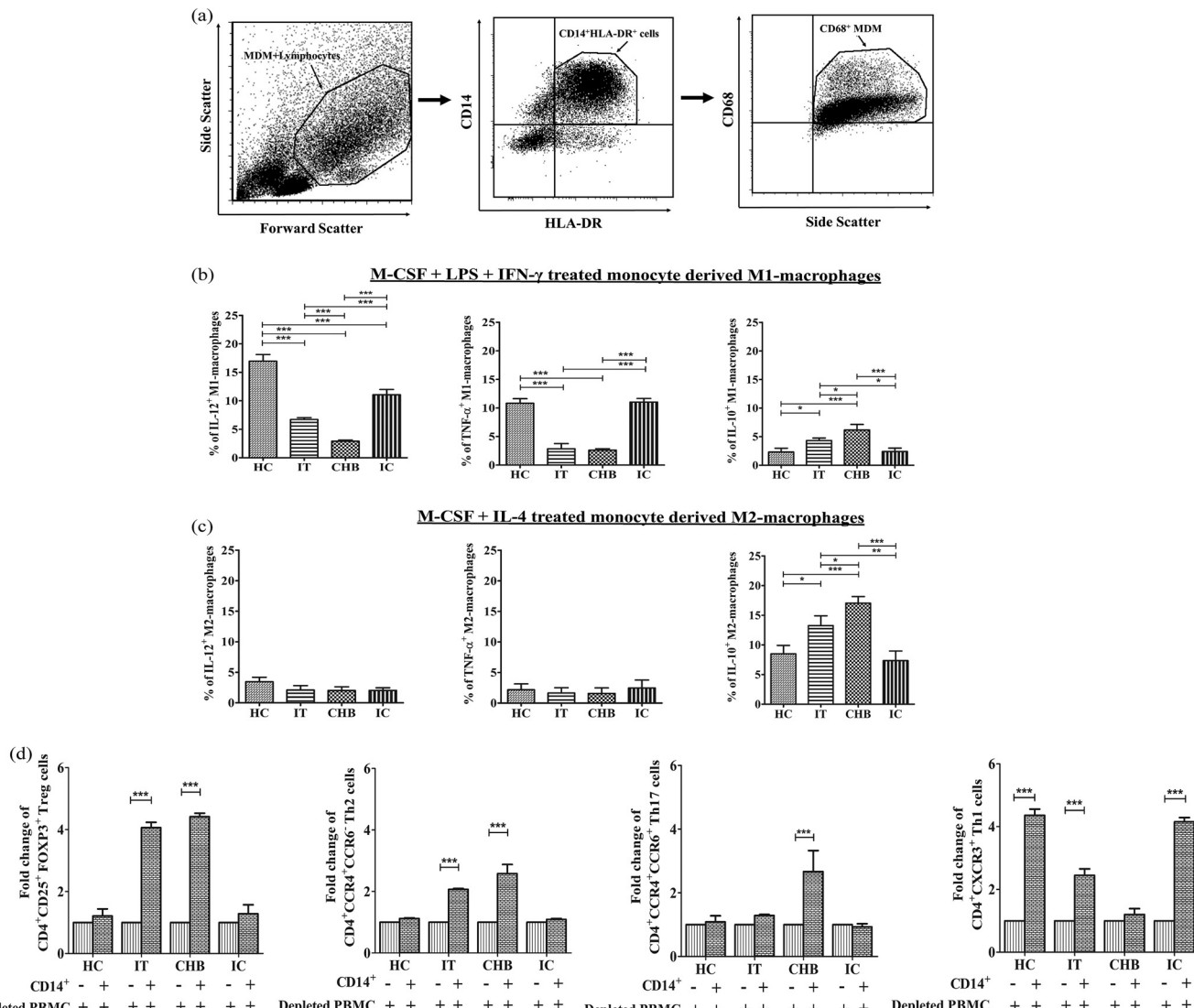

**FIG 6** Cytokine profile of monocyte-derived macrophages and monocyte-mediated CD4$^+$ T-cell differentiation during chronic HBV infection. (a) Sequential gating strategy for the identification of HLA-DR$^+$ CD14$^+$ CD68$^+$ monocyte-derived macrophages (MDMs). (b and c) Percentages (%) of IL-12$^+$, TNF-$\alpha^+$, and IL-10$^+$ M1 macrophages (b) and IL-12$^+$, TNF-$\alpha^+$, and IL-10$^+$ M2 macrophages (c) in different phases of chronic HBV infection and healthy controls (HC). Monocytes were differentiated *in vitro* using M-CSF for 7 days followed by treatment with LPS/rIFN-$\gamma$ for M1 differentiation and IL-4 for M2 differentiation for 24 h. On these differentiated M1 and M2 macrophages, different cytokines were analyzed by intracellular cytokine assay followed by flow cytometric analysis. Means $\pm$ standard deviations are given. Means among groups were compared by one-way ANOVA with Tukey's multiple-comparison test (*, $P < 0.05$; **, $P < 0.005$; ***, $P < 0.0001$). (d) Grouped bar diagram representing fold changes of CD4$^+$ CD25$^+$ FOXP3$^+$ Treg, CD4$^+$ CCR4$^+$ CCR6$^-$ Th2 cells, CD4$^+$ CCR4$^+$ CCR6$^+$ Th17 cells, and CD4$^+$ CXCR3$^+$ Th1 cells after coculture of sorted CD14$^+$ monocytes with anti-CD3/anti-CD28-stimulated autologous monocyte-depleted peripheral blood mononuclear cells (PBMC) for 3 days. For all comparisons, three individual sets of experiments were performed in HC, IT, CHB, and IC, and $P$ value was calculated by the paired Student $t$ test (***, $P < 0.0001$).

compared with IC or HC, and in all cases, the classical monocytes highly expressed CCR2, followed by the intermediate and nonclassical subsets (Fig. S8). To explore whether CCR2 expression on monocytes is regulated by $\beta$-catenin, PBMC of CHB patients were cultured in the presence or absence of a $\beta$-catenin/TCF inhibitor (iCRT3 {2-[[[2-(4-ethylphenyl)-5-methyl-4-oxazolyl]methyl]thio]-$N$-(2 phenylethyl)acetamide}) and CCR2 expression on monocytes was assessed. The percentage of CCR2$^{hi}$ monocytes was found to be significantly reduced upon addition of iCRT3 in comparison to untreated cells, implying that $\beta$-catenin could promote the expression of CCR2 in the monocytes of CHB patients (Fig. 7b).

**Intrahepatic accumulation of $\beta$-catenin$^+$ CD14$^+$ monocytes in CHB patients.** To evaluate the intrahepatic incidence of $\beta$-catenin-expressing monocytes in CHB patients, we studied the frequency of CD14 and $\beta$-catenin double-positive cells in liver biopsy sections of three CHB patients and HC by immunohistochemical staining. Liver histology indicated

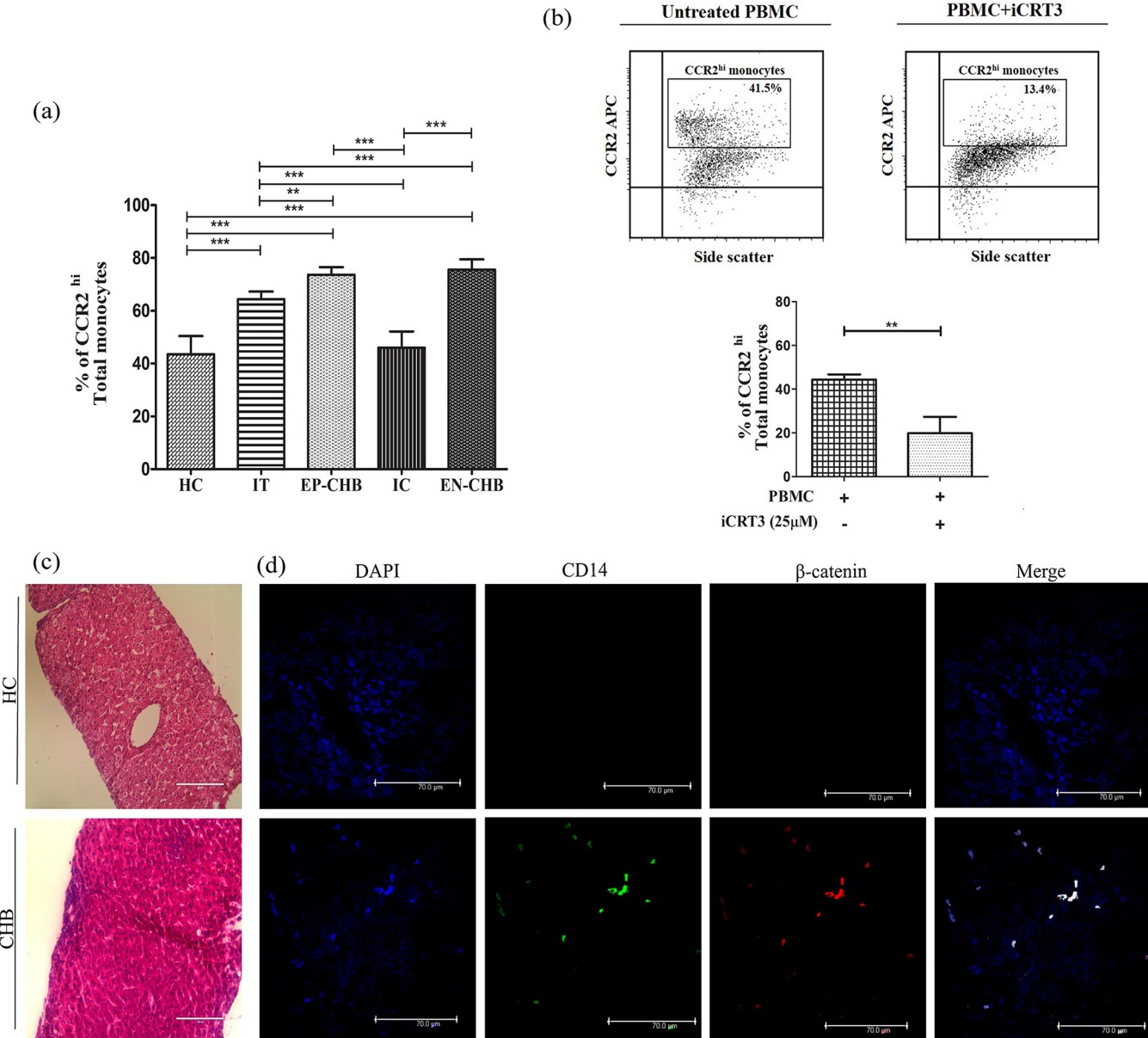

**FIG 7** Homing potential of monocytes and their intrahepatic accumulation during chronic HBV infection. (a) Frequencies of CCR2hi total monocytes in immune tolerance (IT) ($n = 10$), HBeAg-positive CHB (EP-CHB) ($n = 15$), inactive carrier (IC) phase ($n = 22$), HBeAg-negative CHB (EN-CHB) ($n = 18$), and healthy controls (HC) ($n = 19$). Comparisons were done using one-way ANOVA followed by Tukey's multiple-comparison test (**, $P < 0.005$, and ***, $P < 0.0001$). (b) Representative flow cytometric plots (top) and bar diagram (bottom) showing percentages of CCR2hi monocytes in the absence/presence of $\beta$-catenin/TCF inhibitor (iCRT3) (25 $\mu$M). Means $\pm$ standard deviations from three individual sets of experiments are given. Statistical significance was analyzed by paired $t$ test (**, $P < 0.005$). (c) Representative images of hematoxylin and eosin (H&E) staining of liver tissues from HC and CHB patients. (d) Sections were stained with anti-CD14-FITC, anti-$\beta$-catenin-PE, and DAPI. Immunofluorescence images at ×63 magnification showing CD14$^+$ monocytes (green), their $\beta$-catenin expression (red), and their colocalization. Representative images of three individual sets of experiments are shown.

that lymphocyte-predominant lobular and portal inflammation was prominent in CHB compared with HC (Fig. 7c). $\beta$-Catenin$^+$ CD14$^+$ cell density was found to be substantially high in the livers of CHB patients, and such cells were barely perceptible in HC (Fig. 7d).

**Frequency/phenotype/function of monocytes in tenofovir-treated CHB patients.**
Tenofovir is recommended as first-line monotherapy for CHB patients (16), and we tested the effect of tenofovir treatment on the frequency and expression of different functional markers of monocytes in 12 CHB patients who included 5 EP-CHB and 7 EN-CHB patients. We observed that all patients achieved <250 copies/mL of HBV DNA and normalization of serum alanine transaminase (ALT) after 1 year of therapy (Fig. 8a), but no significant change was detected in the monocyte subset distribution or the proportion of TLR-2-, IL-12-, IL-10-,

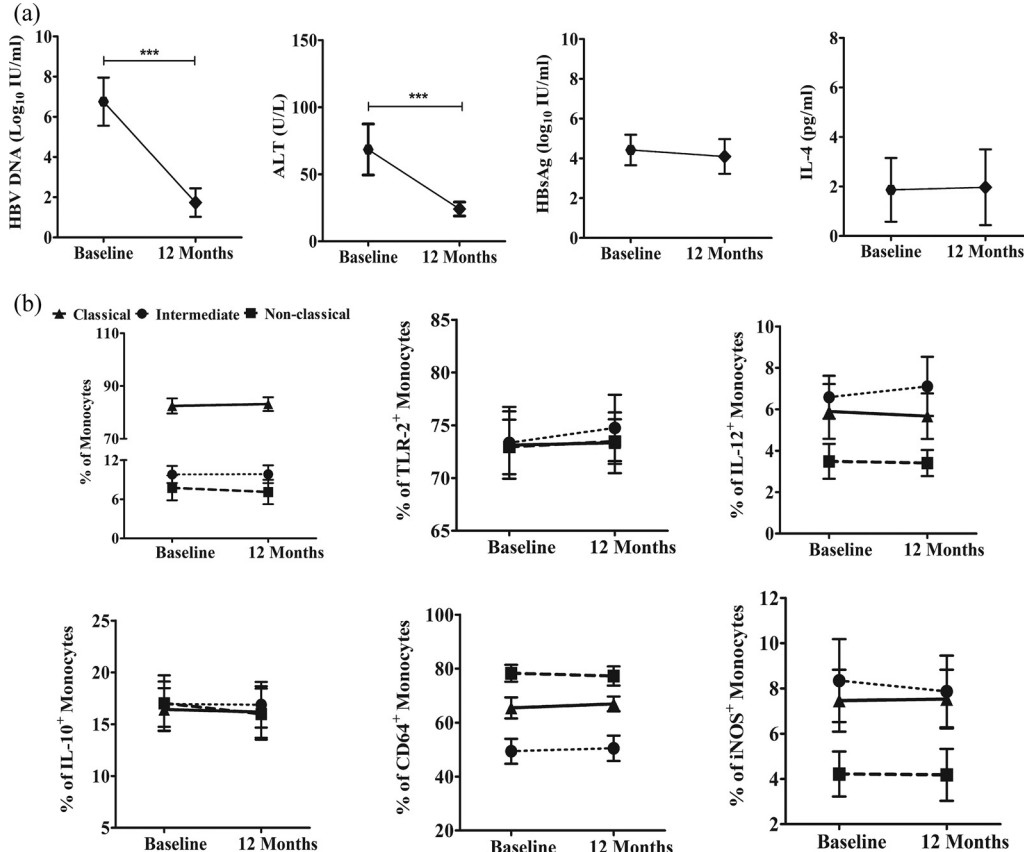

**FIG 8** Monocyte frequencies, functions, and clinical parameters in tenofovir-treated CHB patients. (a) Serum HBV DNA load, serum ALT levels, and serum HBsAg and IL-4 levels. (b) Proportions of monocyte subsets of CHB patients at baseline and at the end of 12 months of tenofovir therapy and percentages of TLR-2-, IL-12-, IL-10-, CD64-, and iNOS-expressing monocyte subsets in CHB patients ($n = 12$) before and after tenofovir treatment. Paired Student's $t$ test and repeated-measures ANOVA were performed for statistical analysis (***, $P < 0.001$).

CD64-, and iNOS-expressing monocyte subsets between pre- and posttreatment time points (Fig. 8b). Moreover, the serum levels of HBsAg and IL-4 in these patients remained similar to baseline values (Fig. 8a).

## DISCUSSION

In the present study, we enumerated the broad spectrum of phenotypic and functional alterations in the monocyte population in chronically HBV-infected patients representing different disease phases, identified the viral antigens and systemic cytokines that induced these phenomena, and studied the mechanisms underpinning the changes. These findings add to the ongoing efforts of defining the process behind immune dysregulation associated with CHI and provide targets for development of new therapies aimed at reversing the defects.

An expansion of the CD16[+] compartment (intermediate and nonclassical monocytes) along with a concomitant decrease in classical monocytes was perceived in IT and CHB patients relative to IC/HC. Similar shifts in monocyte subset distribution had been previously reported in Chinese HBeAg-positive CHB patients (17) and also in other infectious diseases (7, 8). The interactions of monocytes with the pathogen, pathogen-derived factors, or local cytokines were suggested to be the causal factor for this increase in CD16[+] monocytes (18). The secretory HBsAg had been shown to have an immunomodulatory effect, and the circulating CD14[+] monocytes were found to harbor a detectable reservoir of HBsAg in CHB patients (19). We demonstrated that exposure to high HBsAg concentrations stimulated the preferential generation of intermediate and nonclassical monocytes and that this concurred with the higher frequency of these two subsets in IT and CHB patients, who, unlike IC, carried high serum HBsAg levels. A study by Tsai et al. had reported that the sialyl

glycans on HBsAg bind to the immune checkpoint receptor Siglec-3 (CD33) to modulate host immunity (20). The expression of CD33 was found to be significantly enhanced on the monocytes of chronically HBV-infected patients compared to healthy controls (see Fig. S9 in the supplemental material), and it appears that HBsAg might exert its effect on monocytes via its interaction with CD33.

Monocytes are one of the first types of immune cells to come in contact with viral pathogens, and this is in part due to their TLRs that can recognize a number of different viral ligands. Generally, TLR-2 and TLR-4 sense the presence of viral proteins, whereas TLR-7 and -8 bind single-stranded viral RNA and TLR-9 recognizes viral CpG DNA (4). Previous studies on the expression of TLRs on monocytes of CHB patients had revealed in many cases divergent data. Song et al. reported an overexpression of TLR-2 on $CD14^+$ monocytes in HBV-infected Chinese patients compared to HC (21), while Deng et al. reported impaired expression and function of TLR-8 in monocytes (22). In another study conducted in Australia, Visvanathan et al. noted a marked reduction in TLR-2 but not TLR-4 expression on monocytes in patients with HBeAg-positive CHB relative to HBeAg-negative CHB and controls (9). In contrast, we observed that both TLR-2 and TLR-4 along with TLR-9 were downregulated in both EP- and EN-CHB and also in IT in comparison to IC/HC, while we have found no significant differences in TLR-8 expression among the study subjects. These discrepancies in TLR expression profile might partially result from investigating ethnically different patient populations or might be impacted by high exposure of a population to other pathogens causing tuberculosis, malaria, or leprosy or to parasitic worms in countries of endemicity. TLR signaling pathways usually induce the expression of an array of inflammatory cytokines. We noted that the attenuation of TLR expression in CHB and IT was reciprocated by greatly impaired production of TNF-$\alpha$, IL-12, and IL-6 by the monocytes while the levels of IL-10 and TGF-$\beta$ were significantly enhanced. Intriguingly, higher expression of both TNF-$\alpha$ and IL-10 transcripts had been documented in the monocytes of Chinese CHB patients than in those of healthy donors (21). Different studies have established that IL-12 produced by monocytes induces IFN-$\gamma$ production from T cells and also skews the naive $CD4^+$ T cells toward the Th1 phenotype (23), which helps in eliminating viral infection. Conversely, upregulation of IL-10 in monocytes coincides with impaired T-cell responses directed against the virus (23). Hence, it appears that the inhibition of IL-12 and augmentation of IL-10 by monocytes in CHB/IT affect the antiviral activities of T-cells and favor viral persistence.

A salient feature of the monocytes is their capability to phagocytose foreign organisms and generate ROS and RNS by specialized enzymes, NADPH oxidase and iNOS, which can irreversibly oxidize and damage the cellular structures of the intruding pathogens.

A significant reduction in phagocytic activity and intracellular ROS production along with suppression of iNOS was manifested by the monocytes of EP-/EN-CHB and IT compared with IC/HC. Our finding was in accord with that of Prieto et al., who had earlier reported diminished phagocytosis by monocytes of CHB patients relative to HBeAg-negative and anti-HBe-positive patients and healthy subjects (24). This overall inadequate antimicrobial activity of monocytes in CHB and IT contributes to the chronicity of infection and high viral load in these phases.

Monocytes give rise to macrophages in tissues, and the heterogeneity in monocytes underlies that of macrophages. To determine if MDM subsets are functionally altered in CHI, we investigated the cytokine profiles of these macrophages, which can confer a better appreciation of functional polarization than cell surface receptor expression. Our results revealed that M1 macrophages from CHB and IT acquired an M2-like anti-inflammatory state characterized by decreased production of TNF-$\alpha$ and IL-12 and an increase in IL-10. Similar cytokine secretion features of M1 cells had been reported in chronic HCV patients (8). However, unlike M2 subsets from HCV-infected individuals, which secrete more proinflammatory cytokines than controls (8), the M2 macrophages in all phases of CHI produced very low concentrations of IL-12 and TNF-$\alpha$. Moreover, M2 cells generated from CHB and IT displayed remarkable enhancement in IL-10 production in comparison to IC/HC. Thus, the aberrant functions of MDMs in CHB and IT and their shifts toward the M2 phenotype are likely to influence T-cell activation and function that would contribute to viral perpetuation and pathogenicity in CHI.

We next attempted to identify the HBV antigen and host cytokines that might dictate the functional modification of monocytes during CHI. We demonstrated that the high concentration of HBsAg encountered in IT and EP-/EN-CHB potentiated the reduced expression of TLR-2, CD64, iNOS, and IL-12 and heightened expression of IL-10 by the monocytes. Moreover, HBsAg, even at low concentrations, could inhibit IL-12 and iNOS expression and thus could explain the observed decreased production of both of these molecules by monocytes in IC relative to HC. The role of HBsAg in selectively inhibiting TLR-2 ligand-induced IL-12 production in monocytes/macrophages had also been previously highlighted by Wang et al. (10). In addition, *in vitro* assays depicted that IL-4, detected at high levels exclusively in EP-/EN-CHB phase, could constrain IL-12 but stimulate IL-10 expression in monocytes and thus could account for the greater immunosuppressor trait of monocytes in CHB patients. Similar IL-4-mediated inhibition of IL-12 production had been observed in murine peritoneal macrophages (25), and IL-4 had also been implicated in promoting IL-10 production in CD4$^+$ T cells (26).

We dissected the molecular mechanisms through which HBsAg and IL-4 could affect the phenotype and function of monocytes and showed that both pathways impinge on the activation of $\beta$-catenin. An increased accumulation of $\beta$-catenin was seen in HBsAg- or IL-4-treated monocytes only at the protein level but not at the mRNA level. A similar increase in $\beta$-catenin protein but not mRNA amount in mouse macrophages following exposure to IL-4 had been previously reported (27). The contrary effect of HBsAg on $\beta$-catenin mRNA was relatively small in magnitude and presumably eclipsed by its far greater effect on $\beta$-catenin protein. Pharmacological inhibition of $\beta$-catenin could potentially normalize the immune functions in these cells, signifying a crucial regulatory role of $\beta$-catenin in monocyte function during CHI. We also observed a positive correlation between HBsAg level and frequency of $\beta$-catenin$^+$ monocytes in chronically HBV-infected patients. It seems plausible that a high or low $\beta$-catenin concentration results in distinct effects on the expression of different functional markers of monocytes and thus could account for the difference in monocyte properties in CHB and IC.

The monocytes in CHB and IT displayed enhanced CCR2 expression and thus appeared to have a high propensity to migrate to the inflamed liver. Interestingly, in regorafenib-resistant cancer cells, $\beta$-catenin was found to be a direct transcriptional activator of CCR2 expression (15). We also noticed that inhibition of $\beta$-catenin in monocytes of CHB patients resulted in significant reduction of CCR2 expression by these cells, supporting the idea that $\beta$-catenin is involved in the induction of CCR2 expression on monocytes. Further, a greater prevalence of $\beta$-catenin-expressing monocytes was detected in liver tissues of CHB patients than in HC. Intrahepatic monocytes were also previously reported to be expanded in human liver disease (28).

Apart from the innate effector functions, monocytes can also act as a bridge to the adaptive immune system. Colocalization of CD14$^+$ cells with clusters of CD4$^+$ T cells had been reported at sites of inflammation (6), suggesting an interaction between these cells *in vivo*. We showed that monocytes sorted from CHB shifted the CD4$^+$ T-cell differentiation more toward Treg development, although frequencies of Th2 and Th17 cells were also enhanced. However, there was no monocyte-dependent enrichment of Th1 cells in CHB. On the other hand, in IT, the monocytes dictated the expansion of CD4$^+$ T cells mostly toward Treg and Th2 cells but impeded Th17 development and instead favored the formation of Th1 cells. In contrast, the CD4$^+$ T cells were induced to differentiate mostly toward the Th1 phenotype by the monocytes of IC and HC. Thus, monocytes are differentially programmed in different phases of CHI and contribute to the development of a distinct CD4$^+$ T-cell subpopulation that in turn exerts a profound effect on disease chronicity and pathogenicity.

Finally, we tested whether the potent antiviral drug tenofovir could modulate the monocyte phenotype and function in CHB patients. Our results highlighted that the monocyte subsets continued to retain their baseline characteristics after 1 year of tenofovir therapy, and no reduction in HBsAg or IL-4 level was perceived. The persistence of an immune-suppressive cascade in CHB patients even after effective tenofovir treatment may represent a pivotal risk factor for the advancement of liver diseases.

Taken together, this study has illustrated a multitude of functional defects in monocyte subsets in different phases of CHI. It thus appears that therapeutic targeting of intramonocytic $\beta$-catenin or reducing the circulating HBsAg levels or modulation of the cytokine milieu in chronically HBV-infected patients might be successful in restoring the monocyte function, clearance of HBV, and cure of HBV-induced liver disease.

## MATERIALS AND METHODS

**Study subjects and samples.** Treatment-naive, chronically HBV-infected individuals having hepatitis B surface antigen (HBsAg) positivity for more than 6 months were enrolled from the Hepatology Clinic of the School of Digestive and Liver Diseases (SDLD), Institute of Post Graduate Medical Education and Research (I.P.G.M.E.&R.), Kolkata, India. Necessary virological, biochemical, and clinical assessments were conducted for precisely categorizing the patients into IT, EP-CHB, IC, and EN-CHB phases.

The IT group included participants having HBeAg positivity and high HBV DNA ($>10^6$ copies/mL) but normal serum alanine transaminase (ALT) levels ($\leq$40 IU/L) in three consecutive follow-ups within a period of 1 year prior to recruitment and having minimal or no liver necroinflammation. Both EP-CHB and EN-CHB patients were found to have HBV DNA ($>10^4$ copies/mL) and showed increased ALT levels ($>$40 IU/L) and evidence of active hepatic necroinflammatory activity, the only difference being the presence/absence of detectable HBeAg in their serum.

IC patients were characterized by HBeAg negativity, HBV DNA ($<10^4$ copies/mL), and normal ALT levels ($\leq$40 IU/L) on three follow-ups 3 months apart, with no biochemical/clinical/histological evidence of liver disease.

Patients having coinfection with HIV/HCV/HDV, significant comorbidities like diabetes mellitus, chronic alcoholism, intravenous drug abuse, or evidence of any carcinoma, overt infection, or autoimmune disorders were excluded from the study.

Additionally, HBV-uninfected healthy control subjects without any viral/bacterial infection and chronic/acute illness in the preceding 6 months were included in the study.

Blood samples were drawn from all participants for performing immunological or biochemical assessments. Percutaneous liver biopsy samples were collected from selected CHB patients as deemed necessary by the clinician for studying the liver histopathology. Liver tissues from healthy donors were obtained as part of the pretransplantation assessment for living-donor liver transplantation. Written informed consent was obtained from the participants or from parents or legal guardians of minor participants prior to study inclusion. The access to human samples and all experimental protocols were carried out in accordance with the approved guidelines of the Ethical Review Committee of I.P.G.M.E.&R.

**Reagents and chemicals.** Fluorochrome-conjugated anti-human monoclonal antibodies (MAbs), immunological solutions and buffers, and recombinant proteins were purchased from BD Biosciences (San Jose, CA, USA), BD Pharmingen (San Diego, CA, USA), BioLegend (San Diego, CA, USA), Invitrogen (Grand Island, NY, USA), and Santa Cruz Biotechnology (USA). LPS, brefeldin A, phorbol myristate acetate (PMA), 2',7'-dichlorodihydrofluorescein diacetate (DCFH-DA), $\beta$-galactosidase ($\beta$-Gal), $\beta$-catenin/TCF inhibitor (iCRT3), bovine serum albumin (BSA), and 4',6-diamidino-2-phenylindole (DAPI)/antifade solution were procured from Sigma-Aldrich (St. Louis, MO, USA). Recombinant HBsAg (adw) was obtained from Abcam (Cambridge, England, UK) and RPMI-1640 medium and heat-inactivated fetal bovine serum (FBS) from Gibco (Gaithersburg, MD, USA). An AutoMACS separation unit and MiniMACS starting kit along with anti-CD14 MicroBeads were procured from Miltenyi Biotec (Bergisch Gladbach, Germany). All information about the reagents and chemicals is given in the supplemental material (see Table S2).

**Characterization of circulating total monocytes and monocyte subsets.** The frequency, subsets, and phenotypes of monocytes were determined by staining the freshly collected EDTA-blood of the study subjects with fluorochrome-conjugated monoclonal antibodies (MAbs), namely, anti-HLA-DR-v500, anti-CD14-FITC, anti-CD16-phycoerythrin (PE)-Cy7, anti-TLR-2-PE, anti-TLR-4-allophycocyanin (APC), anti-CD64-PE, anti-LAP-TGF-$\beta$1-APC, and anti-CCR2-APC in appropriate combinations. After 20 min of incubation, red blood cells were lysed with BD fluorescence-activated cell sorting (FACS) lysing solution (BD Biosciences), and the cells were washed, acquired on a FACSVerse (BD Biosciences), and analyzed with FCS Express De Novo software.

To assess the intracellular cytokines and enzyme and TLR expression of the monocytes, EDTA-mixed whole blood was first stimulated with LPS (100 ng/mL) for 1 h followed by incubation with brefeldin A (100 ng/mL) for 3 h. The cells were then stained with anti-HLA-DR-v500, anti-CD14-FITC, and anti-CD16-PE-Cy7 followed by erythrocyte lysis using BD FACS lysing solution. The cells were fixed and permeabilized with the Cytofix/Cytoperm kit (BD Biosciences) and stained with anti-TLR-8-PE, anti-TLR-9-APC, anti-IL-12-PE, anti-IL-6-PE, anti-TNF-$\alpha$-APC, anti-IL-10-APC, anti-iNOS-APC, and anti-$\beta$-catenin-PE in suitable combinations followed by washing and acquisition in a flow cytometer. For flow cytometric analyses, in each set, a fluorescence minus one (FMO) control was used to minimize background noise.

The phagocytic activity of monocytes was assessed using commercially available FITC-conjugated zymosan particles from yeast (Sigma-Aldrich, gifted by Arindam Bhattacharya, Department of Zoology, University of Kolkata). Briefly, 100 $\mu$L of fresh EDTA-blood was incubated with FITC-conjugated zymosan particles for 30 min at 37°C. The phagocytosis was stopped by transferring the samples to ice. The samples were stained with anti-HLA-DR-v500, anti-CD14-peridinin chlorophyll protein (PerCP), and anti-CD16-PE-Cy7 followed by lysis of the erythrocytes, washing of the cells, and analysis by flow cytometry. The percentages of zymosan-FITC-positive monocytes were determined in different study groups.

To determine the generation of ROS by monocytes, 100 $\mu$L of EDTA-blood was first incubated for 20 min with antibodies against HLA-DR, CD14, and CD16. The erythrocytes were lysed as before, and after washing, the cell pellet was suspended in phosphate-buffered saline (PBS). The cells were then treated

with 100 ng/mL PMA for 15 min to stimulate the oxidative burst followed by incubation with 0.01 mM DCFH-DA for 30 min. The fluorescence of oxidized DCF was measured by flow cytometry for determining the intracellular ROS of monocytes.

**Gating strategy of monocytes.** To detect the monocyte population in flow cytometric analysis, first the monocytes together with adjacent lymphocytes were gated by their forward and side scattering property. Next, out of the gated population, CD14$^+$ HLA-DR$^+$ total monocytes were selected on the basis of CD14 and HLA-DR expression. Further, the total monocyte fraction was divided into 3 subtypes depending upon the differential expression pattern of CD14 and CD16, namely, the HLA-DR$^+$ CD14$^{++}$ CD16$^-$ (classical), HLA-DR$^+$ CD14$^{++}$ CD16$^+$ (intermediate), and HLA-DR$^+$ CD14$^+$ CD16$^{++}$ (nonclassical) monocytes. Each subset was individually gated, and the frequency of a particular subset was noted for each sample, using the formula % of frequency of classical/intermediate/nonclassical monocytes = [(% of cells in classical/intermediate/nonclassical gate) × 100]/% of cells in classical + intermediate + nonclassical gate. Subsequently, on each subset, the expression of different surface and intracellular markers was analyzed.

**Serum HBsAg and cytokine quantification.** Serum HBsAg concentrations in HBV-infected patients were quantified using an Abbott Architect i1000sr platform. The levels of different cytokines in serum samples were measured with the BD CBA human Th1/Th2/Th17 cytokine kit (BD Biosciences).

**Treatment of sorted monocytes or PBMC with rHBsAg, $\beta$-catenin/TCF inhibitor, rIL-4, and rTNF-$\alpha$.** PBMC were isolated from EDTA-blood of HC by density gradient centrifugation using HiSep LSM1077 (HiMedia Laboratories, Mumbai, Maharashtra, India). Monocytes were sorted from the isolated PBMC with anti-CD14-coated magnetic beads and separated on an AutoMACS separator (cell purity of >95%) (see Fig. S10 in the supplemental material). In separate experimental setups, sorted CD14$^+$ monocytes were cultured in RPMI medium supplemented with 10% heat-inactivated FBS for 48 h in the presence of rHBsAg (10 $\mu$g/mL and 20 $\mu$g/mL), $\beta$-Gal (20 $\mu$g/mL), or a combination of rHBsAg (20 $\mu$g/mL) and iCRT3 (25 $\mu$M) or kept untreated. The cells were then harvested and stained with anti-HLA-DR-v500, anti-CD14-FITC, anti-CD16-PE-Cy7, anti-TLR-2-PE, and anti-CD64-PE followed by fixation, permeabilization, washing, and staining with anti-IL-12-PE, anti-IL-10-APC, anti-iNOS-APC, and anti-$\beta$-catenin-PE as appropriate and analyzed in a flow cytometer.

Additionally, PBMC of HC were cultured in RPMI medium supplemented with 10% FBS for 3 days in the presence of rIL-4 (5 ng/mL and 25 ng/mL) or rTNF-$\alpha$ (50 ng/mL) or a combination of rIL-4 (25 ng/mL) and iCRT3 (25 $\mu$M) or kept untreated. The cells were then stained as before with antibodies against HLA-DR and CD14, fixed, permeabilized, and incubated with anti-IL-12-PE and anti-IL-10-APC for detection of these intramonocytic cytokines by flow cytometry. Similarly, PBMC of CHB patients were treated with iCRT3 (25 $\mu$M) for 48 h or kept untreated, and CCR2 expression on monocytes was determined.

**Determination of $\beta$-catenin mRNA expression by real-time PCR.** CD14$^+$ monocytes sorted from HC were treated with rHBsAg (20 $\mu$g/mL) or IL-4 (25 ng/mL), and total RNA was extracted from these cells with TRIzol reagent. Subsequently, cDNA was prepared by reverse transcription with RevertAid reverse transcriptase enzyme, and the mRNA expression of $\beta$-catenin was measured by real-time PCR using SYBR green master mix (Applied Biosystems) and specific primers (F, 5′-CATTGTTTGTGCAGCTGCT-3′; R, 5′-GAAGTAACTCTGTCAGAGGA-3′). Gene expression was normalized with endogenous 18S rRNA value.

**Differentiation of monocytes to macrophages.** To differentiate monocytes into macrophages, at first 2 × 10$^7$ PBMC were seeded into each well of a 12-well cell culture plate and allowed to adhere in a 5% CO$_2$ incubator at 37℃ for 5 h in serum-free RPMI medium. After 5 h, the serum-free medium was aspirated carefully from the 12-well culture plate, and RPMI medium supplemented with 10% heat-inactivated FBS was slowly poured over the adherent cells. These cells were then cultured in the presence of M-CSF (50 ng/mL; BioLegend) for 7 days. However, at day 3, the existing cell culture medium was replaced with fresh RPMI medium containing FBS and M-CSF as before. At day 7, the cells were exposed to fresh medium containing either IFN-$\gamma$ (20 ng/mL; BioLegend) and LPS (100 ng/mL) or IL-4 (25 ng/mL; BioLegend) for an additional 24 h to polarize them to M1 or M2 macrophages, respectively. Next, the cells were washed and stained with anti-HLA-DR-v500, anti-CD14-PerCP, and anti-CD68-FITC, followed by fixation, permeabilization, and staining with anti-TNF-$\alpha$-APC, anti-IL-12-PE, and anti-IL-10-APC for determining the production of the cytokines by M1 and M2 macrophages. Finally, the cells were acquired on a flow cytometer.

**Coculture of sorted monocytes with monocyte-depleted PBMC and analysis of CD4$^+$ T-cell subset frequency.** CD14$^+$ monocytes were sorted from PBMC of different study subjects using anti-CD14-coated magnetic beads as before, and the autologous monocyte-depleted PBMC (CD14$^-$ cells) were stimulated with anti-CD3/anti-CD28. In one set, the sorted CD14$^+$ monocytes were cocultured with these prestimulated monocyte-depleted PBMC in a 3:1 ratio for 3 days in RPMI medium supplemented with 10% heat-inactivated FBS, and in the other set, the stimulated monocyte-depleted PBMC were cultured alone under a similar condition. After 3 days, the cells were harvested, washed, and stained with anti-CD4-FITC, anti-CXCR3-PE, anti-CCR4-PE, anti-CCR6-APC, and anti-CD25-PE-Cy7 in the proper combination followed by fixation and permeabilization with the True nuclear buffer set (BioLegend) and staining with anti-FOXP3-PE. The frequencies of different CD4$^+$ T-cell subsets, namely, Th1 cells (CD4$^+$ CXCR3$^+$), Th2 cells (CD4$^+$ CCR4$^+$ CCR6$^-$), Th17 cells (CD4$^+$ CCR4$^+$ CCR6$^+$), and Treg (CD4$^+$ CD25$^+$ FOXP3$^+$) were determined by flow cytometry.

**Immunofluorescence and H&E staining.** The incidence of $\beta$-catenin$^+$ CD14$^+$ monocytes in liver biopsy tissue sections of selected CHB patients and HC was analyzed by immunohistochemical staining. Hematoxylin and eosin (H&E) staining was performed with formalin-fixed, paraffin-embedded 6-$\mu$m liver tissue sections to evaluate the histological status of the livers of study subjects, and liver sections were viewed under a light microscope. For immunohistochemical analyses, after deparaffinization, hydration, and antigen retrieval, tissue sections were stained with anti-CD14-FITC and anti-$\beta$-catenin-PE in a 1:500 dilution followed by washing and cover mounting with Pro-Long Gold antifade/DAPI solution. The sections were examined under a confocal microscope (Leica Microsystems, Germany).

**Longitudinal assessment of virological and immunological parameters in CHB patients receiving tenofovir.** Blood samples were collected from 12 CHB patients treated with tenofovir (300 mg daily) before the initiation of therapy (=baseline) and at the end of 12 months of treatment, and the frequency, phenotypes, and functions of monocyte subsets as well as serum HBV DNA, ALT, HBsAg, and cytokine levels were determined.

**Statistical analysis.** Data were expressed as mean values ± standard deviations. Comparison between groups was done by one-way analysis of variance (ANOVA) followed by Tukey's multiple-comparison test. Linear regression was performed for correlation analysis. Paired Student's $t$ tests and repeated-measures ANOVA were used to determine statistical significance in compared responses. Statistical analysis was performed using GraphPad Prism 5 software. For all tests, a $P$ value of $<0.05$ was considered statistically significant.

**Ethics approval and consent to participate.** The study was approved by the Ethical Review Committee of the Institute of Post Graduate Medical Education and Research (I.P.G.M.E.&R.), Kolkata, India. Written informed consent was obtained from each participant or from parents or legal guardians of minors prior to study inclusion.

**Data availability.** All data and materials are included in the article and supplemental material.

## SUPPLEMENTAL MATERIAL

Supplemental material is available online only.

**SUPPLEMENTAL FILE 1**, PDF file, 2.7 MB.

## ACKNOWLEDGMENTS

We gratefully acknowledge all participants who donated blood for the study. We thank the Indian Institute of Liver and Digestive Sciences, Sonarpur, Kolkata, India, for helping us in quantification of hepatitis B virus surface antigen in the sera of chronically HBV-infected patients. We are thankful to the Multidisciplinary Research Unit, I.P.G.M.E.&R., Kolkata, for providing us the flow cytometry facility. We are grateful to Shyamasundaran Kottilil, Institute of Human Virology, University of Maryland School of Medicine, for his valuable suggestions throughout this study.

The study was supported by a grant from the Department of Science & Technology-Department of Science & Engineering Research Board, Ministry of Science & Technology, Government of India (project no. CRG/2018/001730). D. Dey and S. Pal are supported by Research Fellowships from the Department of Science and Technology, Ministry of Science and Technology (DST-INSPIRE) (DST/INSPIRE Fellowship/2016/IF160296), Government of India, and the Indian Council of Medical Research (67/15/2018/IMM-BMS), respectively. S. Bhadra is supported by a Research Fellowship from the Department of Biotechnology, Government of India (DBT/2019/IPGMER/1305). The Multidisciplinary Research Unit, I.P.G.M.E.&R., Kolkata, which provided the flow cytometry unit, is funded by the Department of Health Research, Ministry of Health & Family Welfare, Government of India [no. V.25011/85/2014-HR(Pt.I)].

We have no conflicts of interest to disclose in connection with the article.

S. Datta, A. Chowdhury, and D. Dey were involved in study conceptualization and design. D. Dey, S. Pal, and S. Bhadra executed different assays and performed flow cytometry. R. Ghosh, B. C. Chakraborty, and D. Dey performed immunohistochemistry procedures, and A. Baidya performed the real-time PCR experiments. D. Dey and S. Datta participated in analysis and interpretation of data. S. K. M. Ahammed and A. Chowdhury scrutinized the clinical parameters of the patients and helped in the patient recruitment procedure. S. Datta and D. Dey drafted the manuscript, and S. Banerjee helped in revising it critically. All authors read and approved the final version of the manuscript.

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
