## [Reviewer comments · Microbiology Spectrum]

Microbiology Spectrum

Multifaceted defects in monocytes in different phases of chronic HBV infection: lack of restoration after antiviral therapy

Debangana Dey, Sourina PAL, Bidhan Chakraborty, Ayana Baidya, Soham Bhadra, Ranajoy Ghosh, Soma Banerjee, SK Ahammed, Abhijit Chowdhury, and Simanti Datta

Corresponding Author(s): Simanti Datta, Institute of Post Graduate Medical Education and Research, Kolkata, West Bengal, India

Review Timeline:

Submission Date:	May 24, 2022
Editorial Decision:	August 10, 2022
Revision Received:	October 26, 2022
Accepted:	November 2, 2022

Editor: Meghan Starolis

Reviewer(s): Disclosure of reviewer identity is with reference to reviewer comments included in decision letter(s). The following individuals involved in review of your submission have agreed to reveal their identity: Yongai Liu (Reviewer #1); Elias A. Rahal (Reviewer #2)

Transaction Report:

DOI: <https://doi.org/10.1128/spectrum.01939-22>

August 10, 2022

Prof. Simanti Datta
Institute of Post Graduate Medical Education and Research, Kolkata, West Bengal, India
Centre for Liver Research, School of Digestive and Liver Diseases
244 A J C Bose Road
Kolkata, West Bengal 700020
India

Re: Spectrum01939-22 (Multifaceted defects in monocytes in different phases of chronic HBV infection: lack of restoration after antiviral therapy)

Dear Prof. Simanti Datta:

Thank you for submitting your manuscript to Microbiology Spectrum. The review process has been completed, and the reviewers require modifications before it can be accepted. When submitting the revised version of your paper, please provide (1) point-by-point responses to the issues raised by the reviewers as file type "Response to Reviewers," not in your cover letter, and (2) a PDF file that indicates the changes from the original submission (by highlighting or underlining the changes) as file type "Marked Up Manuscript - For Review Only". Please use this link to submit your revised manuscript - we strongly recommend that you submit your paper within the next 60 days or reach out to me. Detailed instructions on submitting your revised paper are below.

Link Not Available

Sincerely,

Meghan Starolis

Journals Department
Reviewer comments:

Reviewer #1 (Comments for the Author):

In this manuscript, Dey et al analyzed and compared defects or alterations in monocyte phenotypes and functions in four stages of chronic HBV infection which including immune tolerant (IT), HBeAg-positive chronic hepatitis B(EP-CHB), HBeAg-negative chronic hepatitis B(EN-CHB) and inactive carriers (IC). They demonstrated that the percentage of HLA-DR+CD14++CD16- classical monocytes were significantly lower than in IT and EP/EN-CHB than in IC and healthy control (HC), and that the percentage of HLA-DR+CD14++CD16+ intermediate monocyte and HLA-CD14+CD16++ non-classical monocytes were significantly higher in IT and EP/EN-CHB than in IC and HC. They further demonstrated that monocytes expression significantly

lower TLRs including TLR2, TLR4 and TLR9 in EP/EN-CHB and IT than in IC and HC. And consistent with previous reports, their also found that monocytes in CHB and IT secreted fewer pro-inflammatory cytokines including IL-12, TNF- α and IL-6, and secreted more immunosuppressive cytokines include IL-10 and TGF- β than in IC and HC. They further explored the mechanism that caused the monocyte defect in CHB and IT. They demonstrated that HBsAg and IL-4 can induce functional defect in monocyte in CHB and IT via activating β -catenin. And monocytes in CHB and IT expressed higher level of CCR2 than in IC and HC. They further demonstrated that one-year of Tenofovir treatment failed to rescue the monocytes function or reduce HBsAg/IL-4 level in serum.

The author's manuscript is based on previous research and has important clinical implications for improving the treatment of CHB. But there are some concerns, especially with the manuscript written and picture organization. Overall, it is interesting. But some conclusions need further investigation. My comments are following:

- 1 . Although it has been reported that CCR2 transcription in tumor cells is controlled by aberrantly activated β -catenin, the authors still need to verify the mechanism in monocytes from HBV patients, because the signaling pathways in immune cells are likely not same as in tumor cells. At the same time, even if more β -catenin+ monocytes are observed in the liver tissue sections of CHB patients, it cannot be clearly proved that β -catenin can promote the expression of CCR2 in the monocytes of CHB, but it can only prove that the expression of the CCR2 is positively correlation with β -catenin. It can be examined as follows: HBV patient-derived monocytes were treated with β -catenin-specific inhibitors, and then q-PCR was used to detect whether the transcription of CCR2 was significantly down-regulated.
- 2 . The mechanisms that the HBsAg and IL-4 promote the activation of β -catenin should be explored further. According to the literature, a dramatic increase in β -catenin protein but not mRNA amount in mouse macrophages following exposure to IL-4 can be detected[1]. Does this mechanism also exist in monocytes-the precursor cells of macrophages? Regarding the regulation of β -catenin by HBsAg, it has been reported that HBsAg can promote the expression of the LEF-1, a downstream mediator and transcription factor of the Wnt signaling pathway[2]. However, there is no report about HBsAg directly promotes the expression of β -catenin. So, this is the first report about HBsAg can directly promote the expression of β -catenin in immune cells. Since β -catenin is closely related to the initiation and progression of liver cancer and the inhibitory phenotype of macrophages including Kupffer cells, and whether HBsAg has an inhibitory effect on the function of immune cells is doubtful for a long time, so it would be a breakthrough in this field if the mechanism that HBsAg promotes β -catenin activation are clearly demonstrated.
- 3 . Please show a representative flow chart for the identification of the purified monocyte isolated by MACS.
- 4 . The immune effects of HBsAg reported in the literature are not always consistent, and sometimes even dramatically opposed. It has been suggested that HBsAg from different sources may differ in lipid profile of subviral articles[3]. Yeast-produced HBsAg is different from HBsAg produced in mammalian cell lines, which could be different from hepatocyte-like cell lines and likely significantly different from HBsAg produced by infected human hepatocytes[3]. So please provide the exact source of HBsAg, including the Cat.no. to ensure that others can follow your experiment. The origin and Cat.no. of other key reagents including FBS, IL-4, TNF- α , antibody used in flow cytometry, MACS kit, purified anti-CD3, purified anti-CD28, LPS are also needed to be clarified.
- 5 . Please clarify whether the FBS used in all experiments were heat-inactivated, and clarify the method for changing cell culture medium during MDM induction.
- 6 . It would be better if arrows were used in the sequential gating strategy for identification cell subsets to clarify the relationship, including in Fig 1a, Fig 5a.
- 7 . Try to avoid using too complicated and too long sentences in the manuscript, which can easily confuse the reader. Here are just a few typical examples. line 146- line 148:"A marked diminution was also observed in the percentages of total-monocytes (Fig. 1d) as well as all three-subsets (Supplementary Fig. S1b) that expressed TLR-regulated pro-inflammatory cytokines TNF- α , IL-12 and IL-6 in IT and EP-/EN-CHB in comparison to IC or HC."line 173-line 177:"We further evaluated the ability of the monocytes to engulf FITC-Zymosan particles and consistent with decreased CD64 expression, there was also substantial decline in the percentage of monocytes associated with zymosan-reporter signal in case of IT and EP-/EN-CHB, suggestive of poor zymosan uptake by these cells than those of IC and HC"line 182-line 184:"The capacity of the monocytes (including all subsets) to produce ROS was significantly attenuated in IT and EP-/EN-CHB as inferred from decreasing DCF fluorescence in these patients relative to HC and IC" line 186-line 187:"Compared to IC/HC, iNOS+ total and all three monocyte-subsets were reduced in numbers in IT and EP-/EN-CHB"line 312:"TLR-2 and TLR-4 sense the presence of virus via their proteins". The expression of these sentences should be improved. Please avoid the expression like "in the percentage of ", "total-monocyte", "all three-subsets", "in case of IT", "by these cells than those of IC and HC", "as inferred from", "in numbers in IT and EP-/EN-CHB", "of virus via their proteins".
- 8 . Please show the results of CD33 expression in monocyte of CHB patients to support your statement in the discussion.
- 9 . The layout of the charts has some problems. In Fig.1b , it would be better to set the maximum value of the ordinate to 10 in the left panel. The ordinate segmentation in Figure 1b (right panel) is inappropriate because it is very hard to clearly see the trend of monocyte abundance changes with the naked eye for reader. And the separated bar graph should be presented here instead of interleaved bar graph. The statistical significances should be only determined between the same monocytes subset of different phase of CHB and HC. Because the percentage of classical monocyte was much higher than the other two subsets of monocytes, so it is absolutely unnecessary to determine whether the percentage of classical monocytes subset was significantly higher than other two subsets of monocytes in every CHB group or HC. So do the Fig.6a (right panel). The graphical labels should highlight important points, and uncontrolled significance analysis may confuse the reader about what the manuscript is trying to emphasize.
- 10 . The Figure legend should be improved to clearly clarify how the sample were treated and how the results were acquired. Especially in Fig.5b and Fig.5c, the authors didn't say anything about the experiment detail or workflow. Figure legend should be relatively independent from the text, so that reader can understand the experimental process just by looking at the figure and

figure legend. The same problem exists in other Figures.

11 . In Fig.3a , the first panel in the left should be designated as a separate figure, because its data type is different from the other panels. The same problem exists in Fig.3b. The two left panels and the two right panels in Fig.3c should be designated as separate figure. At the same time, the gap between the different panels in Fig.3c is too large. The two left panels and the two right panels in Fig.4a should be designated as separate figure.

12 . In Fig.4C, "Percentages of IL-12+ and IL-10+ monocytes" may be mistaken for "percentages of IL-12+IL-10+ monocytes". The figure legend are usually expressed as: Percentage of IL-12+ monocytes(up panel) and IL-10+ monocytes (down panel) . In Fig.4b , the first panel should be a separate figure because only this panel used special symbols.

13 . There are some grammatical mistakes in figure legend of Fig.5d. "following" should be replaced with "after" and "monocyte depleted" should be replaced with "monocyte-depleted".

14 . Some results about different subset monocyte function in HBV infection has been reported[4-7]. And there are some similar results about function and phenotype of macrophages derived from HBsAg-pretreated monocyte has been reported[8]. Please try to cited them in your text and clarify the innovation of your work. And I noticed that the reference [6] has been cited in your manuscript. But please notice that , in fact , some conclusions in the reference[6] are same with yours , for example, the percentage of CD14^{high} CD16⁺ intermediate subset monocytes and CD14⁺CD16^{high} non-classical subset monocytes is significantly high in IA (EP/EN-CHB) than in HC or IT. And I can't find where Zhang et al suggested that overexpression of TLR-2 and TLR-4 on CD14⁺-monocytes in HBV infected Chinese patients as compared to HC in reference [6]. The word "TLR4" only appeared in the title of the cited reference "Kurt-Jones EA, Popova L, Kwinn L, Haynes LM, Jones LP, et al. (2000) Pattern recognition receptors TLR4 and CD14 mediate response to respiratory syncytial virus. *Nat Immunol* 1: 398-401." in that paper. And the word "TLR2" did not appear throughout the paper at all. So please make sure that all references are cited properly in your manuscript.

15 . The percentage of Th17 subset cells in the experiment of MACS-sorted monocytes cocultured with monocyte-depleted PBMC should be determined.

Reference

1. Binder, F., et al., Interleukin-4-induced β -catenin regulates the conversion of macrophages to multinucleated giant cells. *Mol Immunol*, 2013. 54(2): p. 157-63.
2. Daud, M., et al., Modulation of Wnt signaling pathway by hepatitis B virus. *Arch Virol*, 2017. 162(10): p. 2937-2947.
3. Maini, M.K. and A.J. Gehring, The role of innate immunity in the immunopathology and treatment of HBV infection. *J Hepatol*, 2016. 64(1 Suppl): p. S60-s70.
4. Gehring, A.J., et al., Mobilizing monocytes to cross-present circulating viral antigen in chronic infection. *J Clin Invest*, 2013. 123(9): p. 3766-76.
5. Boltjes, A., et al., Monocytes from chronic HBV patients react in vitro to HBsAg and TLR by producing cytokines irrespective of stage of disease. *PLoS One*, 2014. 9(5): p. e97006.
6. Zhang, J.Y., et al., Hyper-activated pro-inflammatory CD16 monocytes correlate with the severity of liver injury and fibrosis in patients with chronic hepatitis B. *PLoS One*, 2011. 6(3): p. e17484.
7. Liaskou, E., et al., Monocyte subsets in human liver disease show distinct phenotypic and functional characteristics. *Hepatology*, 2013. 57(1): p. 385-98.
8. Faure-Dupuy, S., et al., Hepatitis B virus-induced modulation of liver macrophage function promotes hepatocyte infection. *Journal of Hepatology*, 2019. 71(6): p. 1086-1098.

Reviewer #2 (Comments for the Author):

The paper by Dey et al. examines monocyte subsets in chronic hepatitis B infection patient groups. Findings indicate alterations in cell numbers and responses based on the patient type. This manuscript was an enjoyable read as it is generally well written and thought-out; however, I have the following editorial and experimental comments and suggestions:

1-In figure 1b, statistical comparisons between classic and intermediate or non-classic monocytes is not necessary within the same sample type and rather overcomplicates the figure. What is quite necessary to indicate in the figure are significance comparisons for each monocyte type across sample types and the legend can indicate this. Similarly for 3b and 6b.

2-Were other types of infections and conditions as well as medications that affect monocyte subtypes ruled out?

3-Number of samples and repeats should be clarified per data set.

4-The authors should address why TLR 8-expression was not affected compared to other TLRs tested. Deng et al (*Clin. Res. Hepatol. Gastroenterol.* 2017) have previously reported impaired expression and function of TLR8 in monocytes in chronic HBV infection.

5-Further demographic data about the subjects should be included. For example are subject ages matched across the groups?

6-The Hepatitis D virus as a possible co- or super-infection should be assessed in the subjects. This virus may modify the HBV disease course and may concomitantly affect immune responses.

7-Why are the data points connected by lines across different treatments in Fig 4. B and C? This misleadingly gives the impression of a time course. I am also not sure why the authors use a scatterplot in one figure (Fig 1b) to indicate a continuous variable but a bar graph (ex. Fig 2) or median/mean bars (Fig 4) in others. I suggest unifying the style of the visuals for similar types of data.

8-While some figure legends indicate whether the mean or median along with the type of error represented, other figure legends do not. I suggest adding these to all figure legends.

9-Is there a rationale to not study the homing/mobility and beta catenin expression in the immune-tolerant and inactive carrier groups? Based on the CCR2 data, as well as the other differences combined, it would be relevant to examine homing in these groups as well. On the other hand, how many samples of HC and CHB were examined for homing and how concordant was this across subjects? This should be indicated.

Staff Comments:

Preparing Revision Guidelines

Please return the manuscript within 60 days; if you cannot complete the modification within this time period, please contact me. If you do not wish to modify the manuscript and prefer to submit it to another journal, please notify me of your decision immediately so that the manuscript may be formally withdrawn from consideration by Microbiology Spectrum.

In this manuscript, Dey et al analyzed and compared defects or alterations in monocyte phenotypes and functions in four stages of chronic HBV infection which including immune tolerant (IT), HBeAg-positive chronic hepatitis B(EP-CHB), HBeAg-negative chronic hepatitis B(EN-CHB) and inactive carriers (IC). They demonstrated that the percentage of HLA-DR⁺CD14⁺⁺CD16⁻ classical monocytes were significantly lower than in IT and EP/EN-CHB than in IC and healthy control (HC), and that the percentage of HLA-DR⁺CD14⁺⁺CD16⁺ intermediate monocyte and HLA-CD14⁺CD16⁺⁺ non-classical monocytes were significantly higher in IT and EP/EN-CHB than in IC and HC. They further demonstrated that monocytes expression significantly lower TLRs including TLR2, TLR4 and TLR9 in EP/EN-CHB and IT than in IC and HC. And consistent with previous reports, their also found that monocytes in CHB and IT secreted fewer pro-inflammatory cytokines including IL-12, TNF- α and IL-6, and secreted more immunosuppressive cytokines include IL-10 and TGF- β than in IC and HC. They further explored the mechanism that caused the monocyte defect in CHB and IT. They demonstrated that HBsAg and IL-4 can induce functional defect in monocyte in CHB and IT via activating β -catenin. And monocytes in CHB and IT expressed higher level of CCR2 than in IC and HC. They further demonstrated that one-year of Tenofovir treatment failed to rescue the monocytes function or reduce HBsAg/IL-4 level in serum.

The author's manuscript is based on previous research and has important clinical implications for improving the treatment of CHB. But there are some concerns, especially with the manuscript written and picture organization. Overall, it is interesting. But some conclusions need further investigation. My comments are following:

1. Although it has been reported that CCR2 transcription in tumor cells is controlled by aberrantly activated β -catenin, the authors still need to verify the mechanism in monocytes from HBV patients, because the signaling pathways in immune cells are likely not same as in tumor cells. At the same time, even if more β -catenin⁺ monocytes are observed in the liver tissue sections of CHB patients, it cannot be clearly proved that β -catenin can promote the expression of CCR2 in the monocytes of CHB, but it can only prove that the expression

of the CCR2 is positively correlation with β -catenin. It can be examined as follows: HBV patient-derived monocytes were treated with β -catenin-specific inhibitors, and then q-PCR was used to detect whether the transcription of CCR2 was significantly down-regulated.

2. The mechanisms that the HBsAg and IL-4 promote the activation of β -catenin should be explored further. According to the literature, a dramatic increase in β -catenin protein but not mRNA amount in mouse macrophages following exposure to IL-4 can be detected[1]. Does this mechanism also exist in monocytes-the precursor cells of macrophages? Regarding the regulation of β -catenin by HBsAg, it has been reported that HBsAg can promote the expression of the LEF-1, a downstream mediator and transcription factor of the Wnt signaling pathway[2]. However, there is no report about HBsAg directly promotes the expression of β -catenin. So, this is the first report about HBsAg can directly promote the expression of β -catenin in immune cells. Since β -catenin is closely related to the initiation and progression of liver cancer and the inhibitory phenotype of macrophages including Kupffer cells, and whether HBsAg has an inhibitory effect on the function of immune cells is doubtful for a long time, so it would be a breakthrough in this field if the mechanism that HBsAg promotes β -catenin activation are clearly demonstrated.
3. Please show a representative flow chart for the identification of the purified monocyte isolated by MACS.
4. The immune effects of HBsAg reported in the literature are not always consistent, and sometimes even dramatically opposed. It has been suggested that HBsAg from different sources may differ in lipid profile of subviral articles[3]. Yeast-produced HBsAg is different from HBsAg produced in mammalian cell lines, which could be different from hepatocyte-like cell lines and likely significantly different from HBsAg produced by infected human hepatocytes[3]. So please provide the exact source of HBsAg, including the Cat.no. to ensure that others can follow your experiment. The origin and

Cat.no. of other key reagents including FBS, IL-4, TNF- α , antibody used in flow cytometry, MACS kit, purified anti-CD3, purified anti-CD28, LPS are also needed to be clarified.

5. Please clarify whether the FBS used in all experiments were heat-inactivated, and clarify the method for changing cell culture medium during MDM induction.
6. It would be better if arrows were used in the sequential gating strategy for identification cell subsets to clarify the relationship, including in Fig 1a, Fig 5a.
7. Try to avoid using too complicated and too long sentences in the manuscript, which can easily confuse the reader. Here are just a few typical examples. line 146- line 148: “A marked diminution was also observed in the percentages of total-monocytes (Fig. 1d) as well as all three-subsets (Supplementary Fig. S1b) that expressed TLR-regulated pro-inflammatory cytokines TNF- α , IL-12 and IL-6 in IT and EP-/EN-CHB in comparison to IC or HC.” line 173-line 177: “We further evaluated the ability of the monocytes to engulf FITC-Zymosan particles and consistent with decreased CD64 expression, there was also substantial decline in the percentage of monocytes associated with zymosan-reporter signal in case of IT and EP-/EN-CHB, suggestive of poor zymosan uptake by these cells than those of IC and HC” line 182-line 184: “The capacity of the monocytes (including all subsets) to produce ROS was significantly attenuated in IT and EP-/EN-CHB as inferred from decreasing DCF fluorescence in these patients relative to HC and IC” line 186-line 187: “Compared to IC/HC, iNOS⁺ total and all three monocyte-subsets were reduced in numbers in IT and EP-/EN-CHB” line 312: “TLR-2 and TLR-4 sense the presence of virus via their proteins” . The expression of these sentences should be improved. Please avoid the expression like “in the percentage of ”, “total-monocyte”, “all three-subsets”, “in case of IT”, “by these cells than those of IC and HC”, “as inferred from”, “in numbers

in IT and EP-/EN-CHB”、 “of virus via their proteins” .

8. Please show the results of CD33 expression in monocyte of CHB patients to support your statement in the discussion.
9. The layout of the charts has some problems. In Fig.1b, it would be better to set the maximum value of the ordinate to 10 in the left panel. The ordinate segmentation in Figure 1b (right panel) is inappropriate because it is very hard to clearly see the trend of monocyte abundance changes with the naked eye for reader. And the separated bar graph should be presented here instead of interleaved bar graph. The statistical significances should be only determined between the same monocytes subset of different phase of CHB and HC. Because the percentage of classical monocyte was much higher than the other two subsets of monocytes, so it is absolutely unnecessary to determine whether the percentage of classical monocytes subset was significantly higher than other two subsets of monocytes in every CHB group or HC. So do the Fig.6a (right panel). The graphical labels should highlight important points, and uncontrolled significance analysis may confuse the reader about what the manuscript is trying to emphasize.
10. The Figure legend should be improved to clearly clarify how the sample were treated and how the results were acquired. Especially in Fig.5b and Fig.5c, the authors didn't say anything about the experiment detail or workflow. Figure legend should be relatively independent from the text, so that reader can understand the experimental process just by looking at the figure and figure legend. The same problem exists in other Figures.
11. In Fig.3a, the first panel in the left should be designated as a separate figure, because its data type is different from the other panels. The same problem exists in Fig.3b. The two left panels and the two right panels in Fig.3C should be designated as separate figure. At the same time, the gap between the different panels in Fig.3c is too large. The two left panels and the two right panels in Fig.4a should be designated as separate figure.
12. In Fig.4C, “Percentages of IL-12⁺ and IL-10⁺ monocytes” may be

mistaken for "percentages of IL-12⁺IL-10⁺ monocytes". The figure legend are usually expressed as: Percentage of IL-12⁺ monocytes(up panel) and IL-10⁺ monocytes (down panel) . In Fig.4b, the first panel should be a separate

figure because only this panel used special symbols.

▲ Classical
○ Intermediate
■ Non-classical

13. There are some grammatical mistakes in figure legend of Fig.5d. “following” should be replaced with “after” and “monocyte depleted” should be replaced with “monocyte-depleted” .
14. Some results about different subset monocyte function in HBV infection has been reported[4-7]. And there are some similar results about function and phenotype of macrophages derived from HBsAg-pretreated monocyte has been reported[8]. Please try to cited them in your text and clarify the innovation of your work. And I noticed that the reference [6] has been cited in your manuscript. But please notice that, in fact, some conclusions in the reference[6] are same with yours, for example, the percentage of CD14^{high} CD16⁺ intermediate subset monocytes and CD14⁺CD16^{high} non-classical subset monocytes is significantly high in IA (EP/EN-CHB) than in HC or IT. And I can't find where Zhang et al suggested that overexpression of TLR-2 and TLR-4 on CD14⁺-monocytes in HBV infected Chinese patients as compared to HC in reference [6]. The word “TLR4” only appeared in the title of the cited reference “Kurt-Jones EA, Popova L, Kwinn L, Haynes LM, Jones LP, et al. (2000) Pattern recognition receptors TLR4 and CD14 mediate response to respiratory syncytial virus. Nat Immunol 1: 398–401.” in that paper. And the word “TLR2” did not appear throughout the paper at all. So please make sure that all references are cited properly in your manuscript.
15. The percentage of Th17 subset cells in the experiment of MACS-sorted monocytes cocultured with monocyte-depleted PBMC should be determined.

Reference

1. Binder, F., et al., *Interleukin-4-induced β-catenin regulates the conversion of macrophages to multinucleated giant cells*. Mol Immunol, 2013.

54(2): p. 157-63.

2. Daud, M., et al., *Modulation of Wnt signaling pathway by hepatitis B virus*. Arch Virol, 2017. **162**(10): p. 2937-2947.
3. Maini, M.K. and A.J. Gehring, *The role of innate immunity in the immunopathology and treatment of HBV infection*. J Hepatol, 2016. **64**(1 Suppl): p. S60-s70.
4. Gehring, A.J., et al., *Mobilizing monocytes to cross-present circulating viral antigen in chronic infection*. J Clin Invest, 2013. **123**(9): p. 3766-76.
5. Boltjes, A., et al., *Monocytes from chronic HBV patients react in vitro to HBsAg and TLR by producing cytokines irrespective of stage of disease*. PLoS One, 2014. **9**(5): p. e97006.
6. Zhang, J.Y., et al., *Hyper-activated pro-inflammatory CD16 monocytes correlate with the severity of liver injury and fibrosis in patients with chronic hepatitis B*. PLoS One, 2011. **6**(3): p. e17484.
7. Liaskou, E., et al., *Monocyte subsets in human liver disease show distinct phenotypic and functional characteristics*. Hepatology, 2013. **57**(1): p. 385-98.
8. Faure-Dupuy, S., et al., *Hepatitis B virus-induced modulation of liver macrophage function promotes hepatocyte infection*. Journal of Hepatology, 2019. **71**(6): p. 1086-1098.

“Multifaceted defects in monocytes in different phases of chronic HBV infection: lack of restoration after antiviral therapy” [Spectrum01939-22]

Response to Reviewers' comments

At the outset, we would like to thank all the reviewers for their valuable comments that greatly helped us in improving the manuscript.

Reviewer #1

- **1. Although it has been reported that CCR2 transcription in tumor cells is controlled by aberrantly activated β -catenin, the authors still need to verify the mechanism in monocytes from HBV patients, because the signaling pathways in immune cells are likely not same as in tumor cells. At the same time, even if more β -catenin⁺ monocytes are observed in the liver tissue sections of CHB patients, it cannot be clearly proved that β -catenin can promote the expression of CCR2 in the monocytes of CHB, but it can only prove that the expression of the CCR2 is positively correlation with β -catenin. It can be examined as follows: HBV patient-derived monocytes were treated with β -catenin-specific inhibitors, and then q-PCR was used to detect whether the transcription of CCR2 was significantly down-regulated.**

As per the suggestion of the reviewer, we performed experiments to demonstrate that CCR2 expression on monocytes of CHB patients is regulated by β -catenin. For this, PBMCs of CHB patients were cultured for 48 hours in presence or absence of β -catenin/TCF inhibitor (iCRT3) (25 μ M) and CCR2 expression on CD14⁺HLA-DR⁺ monocytes was evaluated by flow cytometry. It was observed that the percentage of CCR2^{hi} monocytes were significantly reduced upon addition of iCRT3 in comparison to untreated cells, implying that β -catenin could promote the expression of CCR2 in the monocytes of CHB patients. The specific experimental details have been included in the Methods section of the revised manuscript (page no. 25 and lines 545-546) and the results are described in page no. 13 and lines 284-289 and illustrated in Figure 7b.

- **2. The mechanisms that the HBsAg and IL-4 promote the activation of β -catenin should be explored further. According to the literature, a dramatic increase in β -catenin protein but not mRNA amount in mouse macrophages following exposure to IL-4 can be detected[1]. Does this mechanism also exist in monocytes-the precursor cells of macrophages? Regarding the regulation of β -catenin by HBsAg, it has been reported that HBsAg can promote the expression of the LEF-1, a downstream mediator and transcription factor of the Wnt signaling pathway[2]. However, there is no report about HBsAg directly promotes the expression of β -catenin. So, this is the first report about HBsAg can directly promote the expression of β -catenin in immune cells. Since β -catenin is closely related to the initiation and progression of liver cancer and the inhibitory phenotype of macrophages including Kupffer cells, and whether HBsAg has an inhibitory effect on the function of immune cells is doubtful for a long time, so it would be a breakthrough in this field if the mechanism that HBsAg promotes β -catenin activation are clearly demonstrated.**

As per the suggestion of the reviewer, we explored the mechanism of HBsAg and IL-4 mediated β -catenin induction. For this, the β -catenin protein as well as the mRNA expression were examined in monocytes of HC following separate treatment with exogenous HBsAg and IL-4 by flow cytometry and real time PCR respectively. We observed that HBsAg increased β -catenin protein expression on the monocytes by ~ 3.8 fold while a ~ 1.4 fold decline was observed in β -catenin mRNA amounts. This has been indicated in page no. 10-11, lines 219-226 and in Fig. 4e and Supplementary Fig. S6 of the revised manuscript. In parallel, treatment of IL-4 resulted in ~ 2.9 fold increase in β -catenin protein amounts in monocytes, while no change in its mRNA level was noted (given in page no. 11, lines 230-232; Fig. 4h and Supplementary Fig. S7). These observations suggested that the induction of β -catenin by HBsAg and IL-4 was mainly at the level of protein synthesis or stability and not at the transcriptional level.

- **3. Please show a representative flow chart for the identification of the purified monocyte isolated by MACS.**

As suggested by the reviewer, a representative flow chart for the identification of the purified monocytes isolated by MACS has been provided in Supplementary Information (Figure S10). Briefly, PBMCs isolated from the study subjects were incubated with CD14 microbeads followed by washing with PBS containing 0.2% BSA. The cells were then passed through the MS column (Miltenyi Biotec) in a magnetic field. The unbound CD14 negative cells were

eliminated and the microbead bound CD14 positive cells were flushed into the sample collection tube. Next, the cells were washed and stained with fluorochrome conjugated antibodies against CD14 and HLA-DR and acquired on a flow cytometer. The purity of CD14⁺HLA-DR⁺ monocytes was found to be >95%.

- **4. The immune effects of HBsAg reported in the literature are not always consistent, and sometimes even dramatically opposed. It has been suggested that HBsAg from different sources may differ in lipid profile of subviral particles [3]. Yeast-produced HBsAg is different from HBsAg produced in mammalian cell lines, which could be different from hepatocyte-like cell lines and likely significantly different from HBsAg produced by infected human hepatocytes [3]. So please provide the exact source of HBsAg, including the Cat.no. to ensure that others can follow your experiment. The origin and Cat.no. of other key reagents including FBS, IL-4, TNF- α , antibody used in flow cytometry, MACS kit, purified anti-CD3, purified anti-CD28, LPS are also needed to be clarified.**

Taking into consideration the suggestion given by the reviewer, a complete list of reagents used in the study along with their respective source, origin and Cat.no. has been included in the supplementary information as Supplementary Table S2.

- **5. Please clarify whether the FBS used in all experiments were heat-inactivated, and clarify the method for changing cell culture medium during MDM induction.**

In all experiments, heat-inactivated FBS was used for PBMC culture. This has been indicated in Methods section (page no. 22, line no. 474) of the revised manuscript. In addition, the detailed method of MDM induction is provided in page no. 26, lines 557-570 of the manuscript. In brief, to differentiate monocytes into macrophages, at first 2×10^7 PBMC were seeded into each well of 12 well cell culture plate and allowed to adhere in a 5% CO₂ incubator at 37°C for 5 hours in serum free RPMI medium. After 5 hours the serum free media was aspirated carefully from the 12 well culture plate and RPMI medium supplemented with 10% heat inactivated FBS was slowly poured over the adherent cells. These cells were then cultured in presence of M-CSF (50ng/ml, BioLegend) for 7 days. However, at day 3, the existing cell culture medium was replaced with fresh RPMI medium containing FBS and M-CSF as before. At day 7, the cells were exposed to fresh medium containing either IFN- γ (20 ng/ml, BioLegend) and LPS (100 ng/ml) or IL-4 (25 ng/ml, BioLegend) for additional 24 hours to polarize them to M1 or M2-macrophages respectively.

Next, the cells were washed and stained with anti-HLA-DR-v500, anti-CD14-PerCP and anti-CD68-FITC, followed by fixation and permeabilization and staining with anti-TNF- α -APC, anti-IL-12-PE and anti-IL-10-APC for determining the production of the cytokines by M1 and M2 macrophages. Finally, the cells were acquired on flow cytometer.

- **6. It would be better if arrows were used in the sequential gating strategy for identification cell subsets to clarify the relationship, including in Fig 1a, Fig 5a.**

Fig. 5a in the previous manuscript has been redesignated as Fig. 6a in the revised manuscript. As suggested by the reviewer, arrows were inserted in the sequential gating strategies in both Fig. 1a and Fig. 6a for better clarity.

- **7. Try to avoid using too complicated and too long sentences in the manuscript, which can easily confuse the reader. Here are just a few typical examples. line 146-line 148:"A marked diminution was also observed in the percentages of total-monocytes (Fig. 1d) as well as all three-subsets (Supplementary Fig. S1b) that expressed TLR-regulated pro-inflammatory cytokines TNF- α , IL-12 and IL-6 in IT and EP-/EN-CHB in comparison to IC or HC."line 173-line 177:"We further evaluated the ability of the monocytes to engulf FITC-Zymosan particles and consistent with decreased CD64 expression, there was also substantial decline in the percentage of monocytes associated with zymosan-reporter signal in case of IT and EP-/EN-CHB, suggestive of poor zymosan uptake by these cells than those of IC and HC"line 182-line 184:"The capacity of the monocytes (including all subsets) to produce ROS was significantly attenuated in IT and EP-/EN-CHB as inferred from decreasing DCF fluorescence in these patients relative to HC and IC" line 186-line 187:"Compared to IC/HC, iNOS+ total and all three monocyte-subsets were reduced in numbers in IT and EP-/EN-CHB"line 312:"TLR-2 and TLR-4 sense the presence of virus via their proteins". The expression of these sentences should be improved. Please avoid the expression like "in the percentage of ", "total-monocyte", "all three-subsets", "in case of IT", "by these cells than those of IC and HC", "as inferred from", "in numbers in IT and EP-/EN-CHB", "of virus via their proteins".**

We have modified the result section in the revised manuscript including the sentences and terms specified by the reviewer and tried to make it more lucid and easily readable.

- **8. Please show the results of CD33 expression in monocyte of CHB patients to support your statement in the discussion.**

As suggested by the reviewer, we have provided the results of CD33 expression in monocytes of CHB patients in Supplementary Fig. S9 in support of the statement in the

Discussion (page no. 15 and lines 320-323). The expression of CD33 was found to be significantly enhanced on the monocytes of chronically HBV infected patients in all disease phases as compared to healthy controls.

- **9. The layout of the charts has some problems. In Fig.1b, it would be better to set the maximum value of the ordinate to 10 in the left panel. The ordinate segmentation in Figure 1b (right panel) is inappropriate because it is very hard to clearly see the trend of monocyte abundance changes with the naked eye for reader. And the separated bar graph should be presented here instead of interleaved bar graph. The statistical significances should be only determined between the same monocytes subset of different phase of CHB and HC. Because the percentage of classical monocyte was much higher than the other two subsets of monocytes, so it is absolutely unnecessary to determine whether the percentage of classical monocytes subset was significantly higher than other two subsets of monocytes in every CHB group or HC. So do the Fig.6a (right panel). The graphical labels should highlight important points, and uncontrolled significance analysis may confuse the reader about what the manuscript is trying to emphasize.**

In the revised manuscript, taking into consideration the suggestions of the reviewer, the following changes are made:

- (i) The maximum value of the ordinate was set to 10 in Fig. 1b.
- (ii) Fig. 1b (right panel) has been redesignated to Fig. 1c in the revised manuscript. In Fig. 1c, instead of interleaved bar graph, separated bar graphs have been provided for displaying the frequencies of classical, intermediate and nonclassical subsets between the study groups. The statistical significances have been determined only between the same monocyte subset of patients in different disease phases and HC.
- (iii) Fig. 6a (right panel) in the old manuscript has been designated as supplementary Fig. S8 in the revised manuscript and significance analysis has been modified.

- **10. The Figure legend should be improved to clearly clarify how the sample were treated and how the results were acquired. Especially in Fig.5b and Fig.5c, the authors didn't say anything about the experiment detail or workflow. Figure legend should be relatively independent from the text, so that reader can understand the experimental process just by looking at the figure and figure legend. The same problem exists in other Figures.**

All figure legends have been modified in the revised manuscript and experimental details have been included.

- **11. In Fig.3a, the first panel in the left should be designated as a separate figure, because its data type is different from the other panels. The same problem exists in Fig.3b. The two left panels and the two right panels in Fig.3C should be designated as separate figure. At the same time, the gap between the different panels in Fig.3c is too large. The two left panels and the two right panels in Fig.4a should be designated as separate figure.**

As suggested by the reviewer we have made the following changes in Fig. 3.

- (i) Fig. 3a in the old manuscript has been split into Fig. 3a and Fig. 3b.
- (ii) Fig. 3b in the old manuscript has been fragmented into Fig. 3c, 3d, 3e, 3f, 3g and 3h.
- (iii) Fig. 3c in the old manuscript is now designated as 4a, 4b, 4c and 4d.
- (iv) Fig. 4a in the old manuscript has been separated as Fig. 4e-4h. Subsequently, all other figures were renumbered in the revised manuscript.

- **12. In Fig.4C, "Percentages of IL-12+ and IL-10+ monocytes" may be mistaken for "percentages of IL-12+IL-10+ monocytes". The figure legend are usually expressed as: Percentage of IL-12+ monocytes (up panel) and IL-10+ monocytes (down panel) . In Fig.4b, the first panel should be a separate figure because only this panel used special symbols.**

Fig. 4b and Fig. 4c in the old manuscript have been redesignated as Fig. 5a-5h in the revised manuscript to avoid confusion. We have tried to address all the concerns raised by the reviewer with respect to these figures.

- **13. There are some grammatical mistakes in figure legend of Fig.5d. "following" should be replaced with "after" and "monocyte depleted" should be replaced with "monocyte-depleted"**

In the revised manuscript Fig. 5d has been redesignated as Fig. 7d. As suggested by the reviewer, we have replaced the word "following" with "after" and "monocyte depleted" with "monocyte-depleted" in the revised legend of Fig. 7d.

- **14. Some results about different subset monocyte function in HBV infection has been reported[4-7]. And there are some similar results about function and phenotype of macrophages derived from HBsAg-pretreated monocyte has been reported[8]. Please try to cited them in your text and clarify the innovation of your work. And I noticed that the reference [6] has been cited in your manuscript. But please notice that, in fact, some conclusions in the reference[6] are same with yours, for example, the percentage of CD14high CD16+ intermediate subset monocytes and CD14+CD16high non-classical subset monocytes is significantly high in IA (EP/EN-CHB) than in HC or IT. And I can't find where Zhang et al suggested that overexpression of TLR-2 and TLR-4 on CD14+-monocytes in HBV infected Chinese patients as compared to HC in reference [6]. The word "TLR4" only appeared in the title of the cited reference "Kurt-Jones EA, Popova L, Kwinn L, Haynes LM, Jones LP, et al. (2000) Pattern recognition receptors TLR4 and CD14 mediate response to respiratory syncytial virus. Nat Immunol 1: 398-401."**

in that paper. And the word "TLR2" did not appear throughout the paper at all. So please make sure that all references are cited properly in your manuscript.

Taking into consideration the suggestions by the reviewer, we have cited the studies by Gehring *et al.* (Ref. no. 19), Zhang *et al.* (Ref. no. 17) and Liaskou *et al.* (Ref. No. 28) in the revised manuscript.

We thank the reviewer for pointing out the mistake with respect to the reference no. 17 and 21. In the old manuscript, we had mistakenly stated that “Zhang *et al.* reported an overexpression of TLR-2 and TLR-4 on CD14⁺-monocytes in HBV-infected Chinese patients as compared to HC (17)”. This was actually reported by Song *et al.* J Immunol. 202:2266-2275. We have corrected the sentence as “Song *et al.* reported an overexpression of TLR-2 and TLR-4 on CD14⁺-monocytes in HBV-infected Chinese patients as compared to HC (21)” (page no. 16 and lines 329-330).

- **15. The percentage of Th17 subset cells in the experiment of MACS-sorted monocytes cocultured with monocyte-depleted PBMC should be determined.**

As suggested by the reviewer, we have performed co-culture of sorted monocytes with monocyte-depleted PBMC and analyzed the frequency of TH17 cell subset. This has been indicated in the Methods (page no. 26-27 line no. 571-584) and Result (page no. 12-13 line no. 257-270) section of the revised manuscript and in Fig. 6d.

Briefly, CD14⁺ monocytes were sorted from PBMC of patients in different phases of chronic HBV infection viz. IT, EP-/EN-CHB, IC and also in HC using anti-CD14-coated magnetic-beads and the autologous monocyte-depleted PBMC (CD14⁻ cells) was stimulated with anti-CD3/anti-CD28. In one set, the sorted CD14⁺ monocytes were co-cultured with these pre-stimulated monocyte-depleted PBMC in 3:1 ratio for 3 days in RPMI medium supplemented with 10% FBS and in the other set, the stimulated monocyte-depleted PBMC was cultured alone in similar condition. After 3 days, the cells were harvested, washed and stained with anti-CD4-FITC, anti-CCR4-PE and anti-CCR6-APC in proper combination. The frequencies of Th17 cells [CD4⁺CCR4⁺CCR6⁺] were determined by flow-cytometry. Our results suggested that monocytes of only CHB patients resulted in induction of CD4⁺CCR4⁺CCR6⁺ Th17-subset by ~2.7 fold (Fig. 6d).

Response to comments of Reviewer 2:

- **1-In figure 1b, statistical comparisons between classic and intermediate or non-classic monocytes is not necessary within the same sample type and rather overcomplicates the figure. What is quite necessary to indicate in the figure are significance comparisons for each monocyte type across sample types and the legend can indicate this. Similarly, for 3b and 6b.**

As per the suggestion of the reviewer, we have made the following changes:

- (i) Fig. 1b (right panel) has been redesignated as Fig. 1c in the revised manuscript where the statistical comparisons have been determined only between the same monocyte subset of patients in different disease phases and HC and not between the three monocyte subsets (classical/intermediate/non-classical) in each disease phase.
 - (ii) In the revised manuscript, Fig.3b has been redesignated as Fig. 3c-3h while Fig. 6a (right panel) has been moved to supplementary information as Fig. S8. In all cases, the statistical comparisons have been determined only between the same monocyte subset of patients across the disease phases.
- **2-Were other types of infections and conditions as well as medications that affect monocyte subtypes ruled out?**

All chronically HBV-infected patients included in the study did not have co-infection with HIV/HCV/HDV or evidence of any overt infection and all were treatment-naïve. This has been indicated in the Methods section page no. 21 and line numbers 437-440 and 452-454.

- **3-Number of samples and repeats should be clarified per data set.**

We have included the number of samples and repeats for each of the data sets in the figure legends of the revised manuscript.

- **4-The authors should address why TLR 8-expression was not affected compared to other TLRs tested. Deng et al (Clin. Res. Hepatol. Gastroenterol. 2017) have previously reported impaired expression and function of TLR8 in monocytes in chronic HBV infection.**

Deng *et al.* (Clin. Res. Hepatol. Gastroenterol. 2017) had previously reported impaired expression and function of TLR-8 in monocytes of chronically HBV infected Chinese patients. In contrast, we found no significant differences in TLR-8 expression among the study subjects, although we observed that both TLR-2 and TLR-4 along with TLR-9 were downregulated in both EP- and EN-CHB and also in IT in comparison to IC and HC. These discrepancies in TLR-expression profile might partially result from investigating ethnically different patient populations or might be impacted by high exposure of a population to other pathogens causing tuberculosis, malaria, leprosy or to parasitic worms in endemic countries. This has been indicated in the discussion section of the revised manuscript page no. 16 line number 327-339.

- **5-Further demographic data about the subjects should be included. For example, are subject ages matched across the groups?**

The demographic data of the study subjects has been provided in the Supplementary table S1 of the revised manuscript. Given that the immune tolerant (IT) phase represents the classical early phase of infection, the age of the study subjects in this phase ranged from 5-18 years, whereas the patients in the EP-/EN-CHB and IC were older than IT.

- **6-The Hepatitis D virus as a possible co- or super-infection should be assessed in the subjects. This virus may modify the HBV disease course and may concomitantly affect immune responses.**

According to the literature, in India only 0.78% patients were co-infected with HDV (Ramachandran *et al.*, J Glob Infect Dis. 2020 12:197-201), and in Kolkata only 3.3% individuals had HDV infection (Bhattacharyya *et al.*, Indian J Public Health. 1998. 42:108-12), which is substantially low. We had also assessed HDV infection in chronically HBV infected patients and none of them were found to be positive. Therefore, it is highly unlikely that the HDV would concomitantly affect the immune responses in these patients.

- **7-Why are the data points connected by lines across different treatments in Fig 4. B and C? This misleadingly gives the impression of a time course. I am also not sure why the authors use a scatterplot in one figure (Fig 1b) to indicate a continuous variable but a bar graph (ex. Fig 2) or median/mean bars (Fig 4) in others. I suggest unifying the style of the visuals for similar types of data.**

As suggested by the reviewer, we have made following changes in the data representation:

- (i) Fig. 4b and 4c have been redesignated as Fig. 5a-h in the revised manuscript. In all cases, we have changed the dot plots to bar graphs.
 - (ii) In all the figures, we have used bar graphs for the data representation and unified the styles of the figures with same type of data.
- **8-While some figure legends indicate whether the mean or median along with the type of error represented, other figure legends do not. I suggest adding these to all figure legends.**

We have added the mean or median along with the type of error in all figure legends of the revised manuscript.

- **9-Is there a rationale to not study the homing/mobility and beta catenin expression in the immune-tolerant and inactive carrier groups? Based on the CCR2 data, as well as the other differences combined, it would be relevant to examine homing in these groups as well. On the other hand, how many samples of HC and CHB were examined for homing and how concordant was this across subjects? This should be indicated.**

As the liver biopsy is usually not recommended in IT patients that comprises of mostly children or young individuals, we could not study the homing/mobility and beta catenin expression in the liver tissues of IT patients by immunohistochemical analysis. In addition, the percentages of circulating CCR2 expressing monocytes were comparable in IC and HC and hence, the intrahepatic incidence of these monocytes is likely to be similar in these two study groups. So, in this study, we evaluated homing/mobility and beta catenin expression of monocytes in the liver tissues of three CHB patients and three healthy controls and in all cases β -catenin⁺CD14⁺ cell density was found to be substantially high in CHB and such cells were barely perceptible in HC.

November 2, 2022

Prof. Simanti Datta
Institute of Post Graduate Medical Education and Research, Kolkata, West Bengal, India
Centre for Liver Research, School of Digestive and Liver Diseases
244 A J C Bose Road
Kolkata, West Bengal 700020
India

Re: Spectrum01939-22R1 (Multifaceted defects in monocytes in different phases of chronic HBV infection: lack of restoration after antiviral therapy)

Dear Prof. Simanti Datta:

Your manuscript has been accepted, and I am forwarding it to the ASM Journals Department for publication. You will be notified when your proofs are ready to be viewed.

Sincerely,

Meghan Starolis
Editor, Microbiology Spectrum
